# Physics-Informed Weakly Supervised Learning for Interatomic Potentials

**Makoto Takamoto** [1]  **Viktor Zaverkin** [1]  **Mathias Niepert** [1 2]

## Abstract

Machine learning plays an increasingly important role in computational chemistry and materials science, complementing expensive ab initio and first-principles methods. However, machine-learned interatomic potentials (MLIPs) often struggle with generalization and robustness, leading to unphysical energy and force predictions in atomistic simulations. To address this, we propose a physics-informed, weakly supervised training framework for MLIPs. Our method introduces two novel loss functions: one based on Taylor expansions of the potential energy and another enforcing conservative force constraints. This approach enhances accuracy, particularly in low-data regimes, and reduces the reliance on large, expensive training datasets. Extensive experiments across benchmark datasets show up to 2× reductions in energy and force errors for multiple baseline models. Additionally, our method improves the stability of molecular dynamics simulations and facilitates effective fine-tuning of ML foundation models on sparse, high-accuracy ab initio data. Code and scripts to reproduce the experiments are available at https://github.com/nec-research/PICPS-ML4Sci.

## 1. Introduction

Ab initio and first-principles methods are essential for the computational exploration of molecular and material properties in the chemical sciences and engineering (Parrinello, 1997; Carloni et al., 2002; Iftimie et al., 2005). However, commonly employed ab initio and first-principles approaches such as coupled cluster (CC) (Purvis & Bartlett, 1982; Bartlett & Musiał, 2007) and density functional theory (DFT) (Hohenberg & Kohn, 1964; Kohn & Sham, 1965), respectively, require substantial computational resources.

As a result, they are typically limited to small- or medium-sized systems and short simulation times, which constrains their applicability and the accuracy of predicted properties. Classical force fields can extend these length and time scales, providing a computationally efficient alternative to first-principles approaches but often lack accuracy. Machine-learned interatomic potentials (MLIPs) offer a promising compromise between the accuracy of first-principles methods and the efficiency of classical force fields (Smith et al., 2017; Chanussot et al., 2021; Unke et al., 2021; Merchant et al., 2023; Kovács et al., 2023; Batatia et al., 2023).

Despite their promise, MLIPs face significant challenges. In particular, they require training datasets that comprehensively cover both configurational (atomic positions) and compositional (atomic types) spaces, typically generated via molecular dynamics (MD) simulations based on ab initio or first-principles methods. Due to the high computational cost of these simulations, the resulting datasets are often sparse, limiting the ability of MLIPs to generalize to new molecular and material systems.

Active learning has been used to address the challenge of data sparsity (Li et al., 2015; Vandermause et al., 2020; Zaverkin & Kästner, 2021; Zaverkin et al., 2022; van der Oord et al., 2023; Zaverkin et al., 2024b), but it still relies on the generation of a substantial amount of first-principles data, typically using DFT, to train an initial model capable of guiding phase space exploration through extended MD simulations. This motivates the development of methods that can complement active learning by reducing the reliance on expensive data acquisition. Moreover, MLIPs often suffer from limited generalization and robustness during MD simulations, exhibiting sensitivity to outliers and local structural perturbations. These issues stem largely from the insufficient coverage of configurational and compositional spaces in existing datasets and data generation workflows.

**Contributions.** This paper addresses the challenges of training machine-learned interatomic potentials (MLIPs) with limited data by proposing a physics-informed weakly supervised learning (PIWSL) approach. The method enables an accurate prediction of potential energies and atomic forces in systems subjected to local perturbations. The main contributions are: (i) We introduce PIWSL based on fundamental physical principles, such as the conservative nature of

---

[1]NEC Laboratories Europe, Heidelberg, Germany [2]University of Stuttgart, Stuttgart, Germany. Correspondence to: Makoto Takamoto <makoto.takamoto@neclab.eu>.

*Proceedings of the 42nd International Conference on Machine Learning*, Vancouver, Canada. PMLR 267, 2025. Copyright 2025 by the author(s).

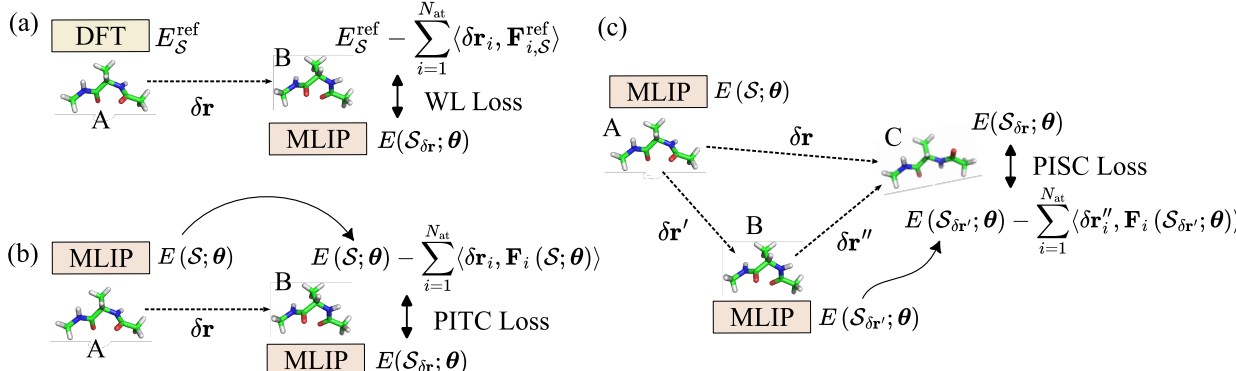

Figure 1: **Schematic illustration of physics-informed weakly supervised losses used in this work.** (a) Taylor-expansion-based weak label (WL) loss with approximate labels obtained from reference energies and atomic forces (Cooper et al., 2020). (b) Physics-inspired Taylor-expansion-consistency (PITC) loss with approximate labels obtained from energies and atomic forces predicted by an MLIP. (c) Physics-inspired spatial consistency (PISC) loss with approximate labels obtained from energies and atomic forces predicted by an MLIP. Here, $E(\mathcal{S}; \boldsymbol{\theta})$ and $\mathbf{F}_i(\mathcal{S}; \boldsymbol{\theta})$ denote the potential energy and atomic forces predicted by an MLIP parametrized by $\boldsymbol{\theta}$, $\mathcal{S}$ and $\mathcal{S}_{\delta\mathbf{r}}$ define the original atomic structure and the one perturbed by $\delta\mathbf{r}$.

forces. By leveraging a Taylor expansion of the potential energy, we derive two novel physics-informed loss functions, physics-informed Taylor consistency (PITC) and physics-informed spatial consistency (PISC), which form the core of the PIWSL framework (illustrated in figure 1b and c). (ii) Through extensive experiments, we demonstrate that PIWSL enables effective training of MLIPs without access to large datasets and enhances robustness during molecular dynamics (MD) simulations. (iii) We show that PIWSL improves the accuracy of total energy and atomic force predictions, even when force labels are unavailable. This is particularly relevant for fine-tuning MLIP foundation models with reference ab initio energies, where force calculations (e.g., at the CCSD(T)/CBS level) are often computationally prohibitive (Smith et al., 2019; 2020; Zaverkin et al., 2023; Hobza & Šponer, 2002; Feller et al., 2006). (iv) Finally, PI-WSL reduces sensitivity issues in MD simulations caused by small training datasets by explicitly modeling the response of the potential energy to local atomic perturbations.

## 2. Related Work

**Machine-Learned Interatomic Potentials.** The field of machine-learned interatomic potentials (MLIPs) emerged over two decades ago (Blank et al., 1995) and has been one of the most active research directions since then (Behler & Parrinello, 2007; Artrith et al., 2011; Artrith & Urban, 2016; Smith et al., 2017; Shapeev, 2016; Schütt et al., 2017; Thomas et al., 2018; Unke & Meuwly, 2019; Drautz, 2019; Zaverkin & Kästner, 2020; Zaverkin et al., 2021; Thomas et al., 2018; Schütt et al., 2021; Shuaibi et al., 2021a; Passaro & Zitnick, 2023; Liao et al., 2023; Batatia et al., 2022; Batzner et al., 2022; Musaelian et al., 2023; Duval et al., 2023; Zaverkin et al., 2024a). The development of local

higher-body-order representations (Shapeev, 2016; Drautz, 2019; Zaverkin & Kästner, 2020; Zaverkin et al., 2021) and the emergence of equivariant message-passing neural networks (MPNNs) (Thomas et al., 2018; Schütt et al., 2021; Shuaibi et al., 2021a; Passaro & Zitnick, 2023; Liao et al., 2023; Batzner et al., 2022; Musaelian et al., 2023; Batatia et al., 2022; Zaverkin et al., 2024a) significantly advanced the field. These methods enable the cost-efficient generation of accurate MLIPs for modeling interactions in many-body atomic systems and account for crucial inductive biases, such as the invariance of the potential energy under rotation. The evaluation of MLIPs for MD simulations has been addressed in previous work (Fu et al., 2023; Bihani et al., 2024).

**Physics-Informed Machine Learning.** Physics-informed machine learning aims to embed known physical principles into the machine learning process. In the context of MLIPs, models based on equivariant message passing neural networks (MPNNs) enforce rotational invariance of the predicted potential energy by design. These models leverage equivariant features to capture and construct many-body interactions in a manner consistent with the symmetries of the underlying physical system (Thomas et al., 2018; Batzner et al., 2022; Batatia et al., 2022; Musaelian et al., 2023; Liao et al., 2023; Zaverkin et al., 2024a). Furthermore, physics constraints can be integrated via auxiliary loss functions, prompting ML models to learn important physical relationships, as demonstrated for physics-informed neural networks (PINNs) (Raissi et al., 2019; Cai et al., 2022), which learn to model solutions of partial differential equations by minimizing residuals during training. The application of physics-informed machine learning to molecular modeling has garnered increasing interest in both the

machine learning and computational chemistry communities (Godwin et al., 2022; Shui et al., 2022; Ni et al., 2024; Liao et al., 2024). In particular, DeNS (Liao et al., 2024), an extension of NoisyNode (Godwin et al., 2022) designed for non-equilibrium structures, shares several similarities with our approach. PIWSL differs from DeNS in two key aspects: (1) PIWSL does not require force labels, allowing it to fine-tune models using only energy data, which is useful for high-accuracy methods like CCSD(T)/CBS; and (2) PIWSL operates without modifying the underlying model architecture, whereas DeNS introduces architectural changes via an additional force module. The potential to combine these two methods to enhance performance remains an open and promising direction for future research. Previous work by Cooper et al. (2020) has provided key motivation for our current research and is discussed in more detail in subsequent sections.

## 3. Background and Problem Definition

**Machine-Learned Interatomic Potentials.** An atomic configuration, denoted as $\mathcal{S} = \{\mathbf{r}_i, Z_i\}_{i=1}^{N_{\mathrm{at}}}$, comprises $N_{\mathrm{at}}$ atoms characterized by their positions $\mathbf{r}_i \in \mathbb{R}^3$ and atomic numbers $Z_i \in \mathbb{N}$. We consider machine-learned interatomic potentials (MLIPs) that map atomic configurations to scalar energies, i.e., $f_{\boldsymbol{\theta}} : \mathcal{S} \mapsto E \in \mathbb{R}$, where $\boldsymbol{\theta}$ denotes trainable parameters. Let $E(\mathcal{S}; \boldsymbol{\theta})$ be the energy predicted by an MLIP for configuration $\mathcal{S}$. In most MLIPs, atomic forces are obtained as the negative gradient of the potential energy with respect to atomic positions:

$$\mathbf{F}_i\left(\mathcal{S}; \boldsymbol{\theta}\right) = -\nabla_{\mathbf{r}_i} E\left(\mathcal{S}; \boldsymbol{\theta}\right).$$

Hence, these MLIPs ensure that the resulting forces are conservative (curl-free) and the total energy is conserved during a dynamic simulation. However, some models are designed to predict atomic forces directly (Hu et al., 2021; Passaro & Zitnick, 2023; Liao et al., 2023; Chanussot et al., 2021). While this approach avoids expensive gradient computations, it does not enforce energy conservation (Chmiela et al., 2017).

Trainable parameters $\boldsymbol{\theta}$ are optimized by minimizing loss functions on training data $\mathcal{D}$ comprising a total of $N_{\mathrm{train}}$ atomic configurations $\{\mathcal{S}^{(k)}\}_{k=1}^{N_{\mathrm{train}}}$ as well as their energies $\{E_{\mathcal{S}}^{\mathrm{ref}}\}_{\mathcal{S} \in \mathcal{D}}$ and atomic forces $\{\{\mathbf{F}_{i,\mathcal{S}}^{\mathrm{ref}}\}_{i=1}^{N_{\mathrm{at}}}\}_{\mathcal{S} \in \mathcal{D}}$:

$$\begin{aligned}
\mathcal{L}\left(\mathcal{D}; \boldsymbol{\theta}\right) &= \sum_{\mathcal{S} \in \mathcal{D}} L\left(S; \boldsymbol{\theta}\right) \\
&= \sum_{\mathcal{S} \in \mathcal{D}} \Big[ C_{\mathrm{e}} \ell\left(E\left(\mathcal{S}; \boldsymbol{\theta}\right), E_{\mathcal{S}}^{\mathrm{ref}}\right) \\
&\qquad + C_{\mathrm{f}} \sum_{i=1}^{N_{\mathrm{at}}} \ell\left(\mathbf{F}_i(\mathcal{S}; \boldsymbol{\theta}), \mathbf{F}_{i,\mathcal{S}}^{\mathrm{ref}}\right) \Big].
\end{aligned} \quad (1)$$

Here, $\ell$ denotes a point-wise loss function such as the absolute and squared error between the predicted and reference total energies and atomic forces. Typically, reference energies $E_{\mathcal{S}}^{\mathrm{ref}}$ and atomic forces $\mathbf{F}_{i,\mathcal{S}}^{\mathrm{ref}}$ are computed with ab initio or first-principles methods such as CC or DFT, respectively. The relative contributions of energies and forces in the loss function of Eq. (1) are balanced by the weighting coefficients $C_{\mathrm{e}}$ and $C_{\mathrm{f}}$.

**Weakly Supervised Learning.** Generating many reference labels with a first-principles approach is challenging due to the high computational cost. Furthermore, the calculation of atomic forces can be infeasible for some high-accuracy ab initio methods, e.g., for CCSD(T)/CBS. In this work, we focus on weakly supervised learning methods to improve the performance of MLIPs in scenarios where only a limited amount of data is available. These involve the generation of approximate but physically motivated total energies for atomic structures generated by small perturbations of their atomic positions, i.e., $\mathcal{S}_{\delta \mathbf{r}} = \{\mathbf{r}_i + \delta \mathbf{r}_i, Z_i\}_{i=1}^{N_{\mathrm{at}}}$ with a perturbation vector $\delta \mathbf{r}$, where $\delta \mathbf{r}_i$ is the perturbation vector for atom $i$. Approximate labels are computed with MLIPs during their training.

## 4. Physics-informed Weakly Supervised Learning

Generating approximate labels for weakly supervised loss functions for MLIPs is particularly challenging. Small perturbations in atomic structures can lead to significant changes in energies and atomic forces, making standard weak supervision techniques, effective in many ML domains (Yang et al., 2022), inapplicable in this context. To overcome this, we propose a physics-informed weakly supervised learning (PIWSL) approach that leverages known physical principles. Specifically, our method incorporates (i) a Taylor expansion of the potential energy to model the system's local response to atomic displacements, and (ii) a spatial consistency loss that encourages consistent energy differences along two distinct spatial paths, composed of three perturbation vectors in total, from a common reference configuration to the same target structure. These two components are combined into the PIWSL loss, which augments the supervised objective with physically grounded weak supervision, improving both robustness and accuracy.

### 4.1. Physics-Informed Taylor-Expansion-Based Consistency Loss

This section introduces the physics-informed Taylor-expansion-based consistency (PITC) loss. Particularly, we relate the energy predicted directly for a displaced atomic configuration with the energy obtained by the Taylor expansion from the original configuration; see figure 1 (b). We estimate the energy for an atomic structure $\mathcal{S}$ drawn from

the training dataset with atomic positions displaced by a vector $\delta\mathbf{r}$: $\mathcal{S}_{\delta\mathbf{r}} = \{\mathbf{r}_i + \delta\mathbf{r}_i, Z_i\}_{i=1}^{N_{\text{at}}}$. For this atomic configuration, we expand the energy predicted by an MLIP in its second-order Taylor series around the atomic perturbation vector $\delta\mathbf{r}_i$ and obtain

$$E\left(\mathcal{S}_{\delta\mathbf{r}}; \boldsymbol{\theta}\right) \approx E\left(\mathcal{S}; \boldsymbol{\theta}\right) - \sum_{i=1}^{N_{\text{at}}} \left[\langle \delta\mathbf{r}_i, \mathbf{F}_i\left(\mathcal{S}; \boldsymbol{\theta}\right)\rangle\right.$$
$$\left. + k_{\text{2nd}}\langle\delta\mathbf{r}_i, \mathbf{F}_i\left(\mathcal{S}_{\delta\mathbf{r}}; \boldsymbol{\theta}\right) - \mathbf{F}_i\left(\mathcal{S}; \boldsymbol{\theta}\right)\rangle\right] + \mathcal{O}\left(\|\delta\mathbf{r}\|^3\right), \quad (2)$$

where $\langle\cdot\rangle$ denotes the inner product, and we used the following relation between forces and gradients of the energy $\mathbf{F}_i \equiv -\nabla_i E$, i.e., defined as the negative gradients of the potential energy. The parameter $k_{\text{2nd}}$ controls the contribution of the second-order term. Setting $k_{\text{2nd}} = 1/2$ recovers the exact second-order Taylor expansion, while $k_{\text{2nd}} = 0$ leads to the first-order approximation[1]. Using approximate labels $E\left(\mathcal{S}_{\delta\mathbf{r}}; \boldsymbol{\theta}\right)$, we define the PITC loss as

$$L_{\text{PITC}}\left(\mathcal{S}; \boldsymbol{\theta}\right) = \ell\Big( E\left(\mathcal{S}_{\delta\mathbf{r}}; \boldsymbol{\theta}\right), E\left(\mathcal{S}; \boldsymbol{\theta}\right)$$
$$-\sum_{i=1}^{N_{\text{at}}}\langle\delta\mathbf{r}_i, (1 - k_{\text{2nd}})\mathbf{F}_i\left(\mathcal{S}; \boldsymbol{\theta}\right) + k_{\text{2nd}}\mathbf{F}_i\left(\mathcal{S}_{\delta\mathbf{r}}; \boldsymbol{\theta}\right)\rangle\Big), \quad (3)$$

where $\ell$ denotes a point-wise loss for regression problems and $\delta r$ is either randomly sampled or determined adversarially; see section 4.4 for more details. Hence, whenever we encounter a structure $\mathcal{S}$ in a batch during training, a new $\delta r$ is computed for each $\mathcal{S}$. Empirically, second-order terms become important when MLIP prediction errors approach the accuracy limit of first-order approximations, i.e., when first-order terms alone no longer suffice to improve accuracy, or when the dataset predominantly consists of relaxed structures.

## 4.2. Physics-Informed Spatial-Consistency Loss

We propose a physics-informed approach for generating weak labels, grounded in the principle of conservative forces. Specifically, we leverage the property that the energy difference between two points on a potential energy surface is independent of the path taken. To construct the weak supervision signal, we consider two distinct paths from a common reference point to the same target point, each composed of a sequence of perturbation vectors. An example of two such paths is illustrated in 1(c). The proposed loss compares the energy obtained by directly displacing the atomic positions of the original configuration $\mathcal{S}$ (denoted as A) by $\delta\mathbf{r}$ (from A to C), with the energy obtained through two consecutive perturbations: $\delta\mathbf{r}'$ (from A to B) followed by $\delta\mathbf{r}''$ (from B to C).

For the first path, we directly predict the energy with an MLIP, i.e., $E\left(\mathcal{S}_{\delta\mathbf{r}}; \boldsymbol{\theta}\right)$, which is related to the approximated

energy at $\mathbf{r} + \delta\mathbf{r}$ using Eq. (3) through the PITC loss. For the second path, we directly compute the energy $E\left(\mathcal{S}_{\delta\mathbf{r}'}; \boldsymbol{\theta}\right)$ for atomic positions displaced by $\delta\mathbf{r}'$, and use it to approximate $E\left(\mathcal{S}_{\delta\mathbf{r}}; \boldsymbol{\theta}\right)$ after applying the second perturbation vector $\delta\mathbf{r}'' \equiv \delta\mathbf{r} - \delta\mathbf{r}'$. The physics-informed spatial consistency (PISC) loss is formally defined as

$$L_{\text{PISC}}\left(\mathcal{S}; \boldsymbol{\theta}\right) = \ell\Big( E\left(\mathcal{S}_{\delta\mathbf{r}}; \boldsymbol{\theta}\right), E_{\text{PITC}}\left(\mathcal{S}_{\delta\mathbf{r}'}, \delta\mathbf{r}''; \boldsymbol{\theta}\right)\Big), \quad (4)$$

where $E_{\text{PITC}}\left(\mathcal{S}_{\delta\mathbf{r}}, \delta\mathbf{r}'; \boldsymbol{\theta}\right)$ is the potential energy estimated via PITC formula in Eq. (2) from the configuration $\mathcal{S}_{\delta\mathbf{r}}$ perturbed by $\delta\mathbf{r}'$. After jointly training with the PITC and PISC losses, the three distinct estimations at $\mathcal{S}_{\delta r}$ become spatially consistent. Importantly, our approach, grounded in the conservative nature of interatomic forces, is not restricted to configurations involving two perturbation paths or three perturbation vectors. We explore several alternative consistency configurations in section D.2.

## 4.3. Combined Physics-Informed Weakly Supervised Loss

Together with the usual MLIP loss function defined in Eq. (1), the overall objective, which we refer to as the PI-WSL loss, can be written as

$$\arg\min_{\boldsymbol{\theta}} \tilde{\mathcal{L}}\left(\mathcal{D}; \boldsymbol{\theta}\right) = \arg\min_{\boldsymbol{\theta}} \sum_{\mathcal{S}\in\mathcal{D}}\left(L\left(\mathcal{S}; \boldsymbol{\theta}\right)\right.$$
$$\left. + C_{\text{PITC}}L_{\text{PITC}}\left(\mathcal{S}; \boldsymbol{\theta}\right) + C_{\text{PISC}}L_{\text{PISC}}\left(\mathcal{S}; \boldsymbol{\theta}\right)\right), \quad (5)$$

where $C_{\text{PITC}}$ and $C_{\text{PISC}}$ are the weights of the weakly supervised PITC and PISC losses.

## 4.4. Perturbation Directions and Magnitudes

The effectiveness of the proposed approach depends on appropriate choices of the perturbation vectors $\delta\mathbf{r}$. We introduce and justify various strategies for generating the perturbations used in Eq. (3) and Eq. (4). Any vector $\delta\mathbf{r}$ can be written as $\delta\mathbf{r} \equiv \epsilon\mathbf{g}/\|\mathbf{g}\|_2$, where $\epsilon$ is the magnitude of $\delta\mathbf{r}$ and $\mathbf{g}/\|\mathbf{g}\|_2$ is its direction. Physical constraints can limit $\epsilon$. Specifically, we can obtain the maximum perturbation length from the validity of the Taylor expansion in Eq. (2), which, as discussed in section 5.3, is typically given as at most 30% of the original bond length whose shortest example is the bond between carbon and hydrogen atoms, about 1.09 Å; see also figure 2 (c) and (d). The specific values of $\epsilon$ chosen for our experiments are provided in section B.1.

To determine $\mathbf{g}/\|\mathbf{g}\|_2$ we explore two strategies. First, we compute it as the unit vector of a perturbation vector sampled from the uniform distribution on the interval $(-1, 1)$ for each direction

$$\delta\mathbf{r}_{\text{rnd}} \equiv \epsilon\mathbf{g}/\|\mathbf{g}\|_2. \quad (6)$$

Second, we compute an adversarial direction following the approach proposed by Goodfellow et al. (2014); Miyato

---

[1]A more detailed derivation is provided in section D.2.2.

et al. (2018). This direction is defined as the gradient of the loss with respect to the atomic coordinates $\mathbf{r}$, indicating the direction in which the prediction error increases most rapidly for the current predicted energy. Assuming an $L_2$-norm constraint on the perturbation magnitude, the adversarial direction can be approximated as in Miyato et al. (2018)

$$\delta\mathbf{r}_{\mathrm{adv}} \equiv \epsilon\mathbf{g}/\|\mathbf{g}\|_2, \text{ where } \mathbf{g} = \nabla_{\mathbf{r}}L_{\mathrm{dist}}(\mathbf{y}^{\mathrm{pred}}, \mathbf{y}^{\mathrm{ref}}), \quad (7)$$

where $L_{\mathrm{dist}}$ denotes a distance-based objective function that is maximized by introducing the adversarial perturbation $\delta\mathbf{r}_{\mathrm{adv}}$ with $\mathbf{y}^{\mathrm{pred}}$ and $\mathbf{y}^{\mathrm{ref}}$ representing the model prediction and reference values, respectively. For computational efficiency, we primarily employ the random perturbation strategy defined in Eq. (6) in our experiments. A quantitative comparison between random and adversarial perturbation directions is presented in section E.

## 5. Experiments

We evaluate our method through extensive experiments designed to address the following objectives: (1) compare PIWSL with established baselines, (2) analyze the effect of PIWSL using the aspirin molecule, including molecular dynamics (MD) simulations, and (3) assess PIWSL's ability to enhance foundation model finetuning on sparse datasets, particularly for energy and force prediction tasks where force labels are unavailable.[2] Unless otherwise noted, our experiments use the first-order PITC. For the MD17 and MD22 datasets, however, the second-order term is incorporated to meet target accuracy requirements.[3]

### 5.1. Models and datasets

We trained the following representative models that are provided in the Open Catalyst code base (Chanussot et al., 2021): SchNet (Schütt et al., 2017), PaiNN (Schütt et al., 2021), SpinConv (Shuaibi et al., 2021a), eSCN (Passaro & Zitnick, 2023), and Equiformer v2 (Liao et al., 2023), covering MLIPs with a smaller (SchNet, SpinConv, PaiNN) and larger number of parameters (eSCN, Equiformer v2). Moreover, we also considered the MACE model (Batatia et al., 2022), a state-of-the-art model that we use to evaluate the impact of PIWSL on the MD17(CCSD) and MD22 dataset. Unless otherwise mentioned and except for SchNet, forces are directly predicted and not computed through the negative gradient of the energy. The results where forces are computed as negative energy gradients are analyzed in section E.1 and section D.9.

---

[2]Additional results are presented in section E, including: (4) a comparison with a prior weakly supervised method, (5) an ablation study, and (6) an analysis of random versus adversarial perturbation vectors.

[3]The impact of the second-order term is discussed in section E.5.

To evaluate the effect and dependency of the physics-informed weakly supervised approach in detail, we performed the training on various datasets: ANI-1x as a heterogeneous molecular dataset (Smith et al., 2020), $TiO_2$ as a dataset for inorganic materials (Artrith & Urban, 2016)[4], the revised MD17 (rMD17) dataset containing small molecules with sampled configurational spaces for each (Chmiela et al., 2017; 2018; Christensen & von Lilienfeld, 2020), the MD22 dataset containing larger molecules (Chmiela et al., 2023), and LMNTO as another material dataset (Cooper et al., 2020); the benchmark results for rMD17, MD22, and LMNTO are provided in section D.1. The detailed description of each dataset is provided in section B.3.

### 5.2. Benchmark Results

We compare models trained using the PIWSL loss (see Eq. (5)) with baseline models trained using the standard supervised loss only (see Eq. (1)). We also compare our approach to a recently proposed data augmentation method that incorporates the task of denoising random perturbations of the atomic coordinates into the learning objective (NoisyNode) (Godwin et al., 2022). A comparison with weak label method (Cooper et al., 2020) is provided in section E. More details on the setup are provided in section B.1. In the following, all evaluation metrics are computed for the test dataset.

**Heterogeneous Molecular Dataset – ANI-1x.** The results provided in Table 1 show that our approach improves the baseline models' performance in almost all cases. In particular, the error reduction for the predicted energies is often between 10 % and more than 50 %. Interestingly, we observe an improved accuracy for potential energies and atomic forces because we include force prediction in PITC and PISC losses, different from the previous work (Cooper et al., 2020). In most cases, except for SchNet, the data augmentation method (NoisyNode) deteriorates the accuracy of the MLIPs because it does not incorporate the proper response of the energy and atomic forces to the perturbation of atomic positions.

**Dependence on Size of Training Data – ANI-1x.** We train MLIPs with training set sizes of $[50, 10^2, 10^3, 10^4, 10^5, 10^6, 5 \times 10^6]$. The results for training data sizes of $10^5$, $10^6$, and $5 \times 10^6$ are provided in section D.1. The results are plotted in figure 2 (a) and (b). Although the observed error reduction depends strongly on the type of MLIP used, the benefit of the weakly supervised losses often decreases slightly with the number of training samples. This result can be expected as the area covered by the weakly supervised losses is also gradually covered by the reference data as the number of training samples increases. Moreover, the gain in

---

[4]A review of solid-state materials datasets for MLIP training can be found in (Lee et al., 2023), for example.

Table 1: **Energy (E) and force (F) root-mean-square errors (RMSEs) for the ANI-1x dataset.** The results are obtained by averaging over three independent runs. Energy RMSE is given in kcal/mol, while force RMSE is in kcal/mol/Å.

| | | $N_{train} = 100$ | | | $N_{train} = 1000$ | | |
| | | Baseline | Noisy Nodes | PIWSL | Baseline | Noisy Nodes | PIWSL |
|---|---|---|---|---|---|---|---|
| SchNet | E | $65.09 \pm 2.42$ | $\mathbf{57.39 \pm 0.05}$ | $60.30 \pm 1.77$ | $31.49 \pm 0.01$ | $\mathbf{31.10 \pm 0.00}$ | $31.50 \pm 0.00$ |
| | F | $29.06 \pm 0.19$ | $\mathbf{25.62 \pm 0.01}$ | $28.20 \pm 0.60$ | $18.94 \pm 0.01$ | $\mathbf{18.10 \pm 0.00}$ | $18.93 \pm 0.00$ |
| PaiNN | E | $168.01 \pm 1.22$ | $464.55 \pm 6.91$ | $\mathbf{109.89 \pm 11.46}$ | $56.62 \pm 2.80$ | $305.76 \pm 33.93$ | $\mathbf{24.53 \pm 0.48}$ |
| | F | $21.33 \pm 0.10$ | $20.82 \pm 0.03$ | $\mathbf{18.76 \pm 0.30}$ | $12.96 \pm 0.06$ | $14.25 \pm 0.18$ | $\mathbf{11.43 \pm 0.05}$ |
| SpinConv | E | $162.14 \pm 7.55$ | $147.73 \pm 2.23$ | $\mathbf{130.97 \pm 8.58}$ | $43.59 \pm 1.71$ | $299.33 \pm 419.10$ | $\mathbf{39.44 \pm 1.31}$ |
| | F | $21.22 \pm 0.43$ | $\mathbf{21.08 \pm 0.43}$ | $21.61 \pm 0.44$ | $14.51 \pm 1.07$ | $15.83 \pm 0.75$ | $\mathbf{13.59 \pm 0.20}$ |
| eSCN | E | $214.52 \pm 7.55$ | $521.92 \pm 12.05$ | $\mathbf{183.70 \pm 9.79}$ | $59.59 \pm 8.92$ | $241.34 \pm 20.16$ | $\mathbf{21.03 \pm 0.56}$ |
| | F | $20.07 \pm 0.27$ | $23.68 \pm 0.11$ | $\mathbf{19.69 \pm 0.05}$ | $12.50 \pm 0.78$ | $14.42 \pm 0.84$ | $\mathbf{11.83 \pm 0.12}$ |
| Equiformer | E | $398.71 \pm 13.69$ | $632.38 \pm 0.11$ | $\mathbf{154.98 \pm 8.83}$ | $54.52 \pm 4.52$ | $854.33 \pm 317.7$ | $\mathbf{20.89 \pm 0.50}$ |
| | F | $20.71 \pm 0.05$ | $21.82 \pm 0.01$ | $\mathbf{20.55 \pm 0.05}$ | $10.10 \pm 0.00$ | $24.79 \pm 2.05$ | $\mathbf{9.68 \pm 0.03}$ |

Table 2: **Energy (F) and force (F) root-mean-square errors (RMSEs) for the TiO$_2$ dataset.** The results are obtained by averaging over three independent runs. Energy RMSE is given in kcal/mol, while force RMSE is in kcal/mol/Å.

| | | $N_{train} = 100$ | | | $N_{train} = 2000$ | | |
| | | Baseline | Noisy Nodes | PIWSL | Baseline | Noisy Nodes | PIWSL |
|---|---|---|---|---|---|---|---|
| SchNet[a] | E | $17.21 \pm 0.00$ | $19.68 \pm 0.00$ | $\mathbf{17.08 \pm 0.00}$ | $10.16 \pm 0.00$ | $44.44 \pm 0.00$ | $\mathbf{10.14 \pm 0.00}$ |
| | F | $2.84 \pm 0.00$ | $\mathbf{2.70 \pm 0.00}$ | $2.83 \pm 0.00$ | $1.87 \pm 0.00$ | $7.45 \pm 0.00$ | $\mathbf{1.85 \pm 0.00}$ |
| PaiNN[b] | E | $14.41 \pm 0.16$ | n/a[b] | $\mathbf{13.95 \pm 0.09}$ | $2.44 \pm 0.03$ | n/a[b] | $\mathbf{2.30 \pm 0.10}$ |
| | F | $1.59 \pm 0.01$ | n/a[b] | $\mathbf{1.56 \pm 0.01}$ | $0.27 \pm 0.02$ | n/a[b] | $\mathbf{0.24 \pm 0.00}$ |
| SpinConv | E | $20.00 \pm 0.42$ | $18.76 \pm 0.74$ | $\mathbf{16.98 \pm 0.99}$ | $2.78 \pm 0.67$ | $2.76 \pm 0.42$ | $\mathbf{2.05 \pm 0.39}$ |
| | F | $1.58 \pm 0.03$ | $\mathbf{1.53 \pm 0.03}$ | $1.59 \pm 0.03$ | $0.67 \pm 0.15$ | $\mathbf{0.61 \pm 0.09}$ | $0.61 \pm 0.05$ |
| eSCN | E | $16.41 \pm 1.10$ | $20.92 \pm 0.00$ | $\mathbf{12.63 \pm 0.78}$ | $1.78 \pm 0.07$ | $20.90 \pm 0.00$ | $\mathbf{0.90 \pm 0.09}$ |
| | F | $1.57 \pm 0.04$ | $1.66 \pm 0.00$ | $\mathbf{1.44 \pm 0.03}$ | $0.42 \pm 0.17$ | $1.66 \pm 0.00$ | $\mathbf{0.16 \pm 0.01}$ |
| Equiformer | E | $18.21 \pm 0.02$ | $19.06 \pm 0.02$ | $\mathbf{13.93 \pm 0.09}$ | $1.73 \pm 0.05$ | $18.54 \pm 0.10$ | $\mathbf{1.27 \pm 0.02}$ |
| | F | $1.56 \pm 0.01$ | $1.64 \pm 0.00$ | $\mathbf{1.51 \pm 0.19}$ | $0.13 \pm 0.01$ | $1.59 \pm 0.00$ | $\mathbf{0.13 \pm 0.00}$ |

[a] We used a larger batch size of 32 for SchNet since we obtained extremely high errors for the batch size of 4. A more detailed discussion of the experimental results for SchNet is provided in section D.1.

[b] Because of a numerical instability of PaiNN when perturbing atomic coordinates, the cutoff radius is reduced from 12 Å to 5 Å in this experiment. Predicted values become n/a when atomic configurations are perturbed.

accuracy of energy predictions is more significant than that for forces trained only indirectly through the consistency constraint in PITC; see Eq. (3). Finally, it is shown that the improvement is more significant for highly parameterized MLIPs, which benefit the most from increasing the training data size through PIWSL.

**Inorganic Bulk Materials – TiO$_2$.** Titanium dioxide (TiO$_2$) is a highly relevant metal oxide for industrial applications, featuring several high-pressure phases. Thus, ML models should be able to predict total energies and atomic forces for various high-pressure phases of TiO$_2$, considering periodic boundaries (relevant when aggregating over the local atomic neighborhood). The results for trained models are provided in Table 2. Similar to the ANI-1x dataset, our approach improves the accuracy of predicted energies and atomic forces. Interestingly, even when the error in the potential energy for 2000 training configurations reaches small RMSE

values, close to or even less than 1 kcal/mol in predicted energy, the PIWSL still reduces the error. This observation indicates strong evidence of the effectiveness of PIWSL applied to bulk materials.

### 5.3. Qualitative Impact of PIWSL

We evaluate the prediction variance and robustness of an MLIP model trained with PIWSL using the aspirin molecule, focusing on the potential energy's dependence on the C–H bond length. In this work, robustness refers to the prediction robustness of an MLIP to perturbations in atomic coordinates. In the literature, the robustness of MLIPs also means their stability during MD simulations. We train PaiNN on the rMD17 aspirin dataset using 100 and 200 configurations with and without the PIWSL loss. The detailed training setup and errors of the used MLIPs are summarized in sec-

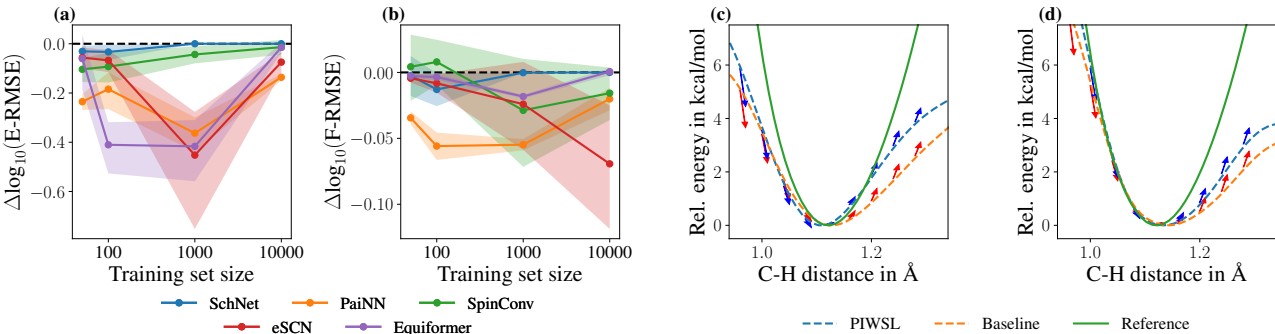

Figure 2: **(a, b) Relative performance gains for MLIPs trained with PIWSL compared to those trained without it and (c, d) potential energy profiles for a C–H bond of the aspirin molecule.** Relative performance gains are evaluated for (a) energy (E-) and (b) force (F-) RMSEs as: $\mathrm{RMSE}_{\mathrm{PIWSL}}/\mathrm{RMSE}_{\mathrm{Baseline}}$. These results are presented for the ANI-1x dataset. Potential energy profiles for a C–H bond of the aspirin molecule are presented for models trained using (c) 100 and (d) 200 configurations. The red and blue arrows indicate the direction from the original structure ($E(\mathcal{S}; \boldsymbol{\theta})$) to the perturbed one ($E(\mathcal{S}_{\delta\mathbf{r}}; \boldsymbol{\theta})$), as defined by Eq. (2), for the baseline and PIWSL model predictions, respectively.

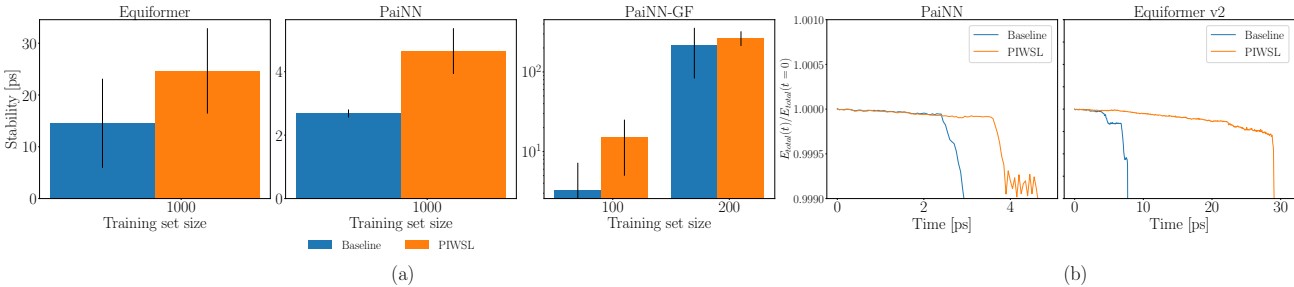

Figure 3: **Stability analysis of the MLIP models during MD simulations.** (a) Stability during MD simulations is assessed for the baseline MLIP models and those trained with PIWSL. Left/Middle: Models with the direct force branch. Right: Models with forces computed as negative gradients of the energy. All results are obtained for the aspirin molecule and MD simulations in the microcanonical (NVE) statistical ensemble. We measure stability during MD simulations according to Fu et al. (2023). (b) Plots of the total energy conservation observed in MD simulations using MLIP models with a force branch trained on 1,000 samples, with and without the proposed loss.

tion D.3.1. We examine the potential energy varying the length of a C–H bond from 0.9 Å to 1.4 Å. The equilibrium C–H bond length is about 1.09 Å. The results in figure 2 (c) and (d) demonstrate that the PIWSL method improves the predicted potential energy profile, indicating improved robustness under perturbations of atom coordinates.

Although the estimated potential energies do not always match the reference values, the direction between the original and perturbed configurations, indicated by arrows in figure 2, consistently follows the gradient of the reference potential energy, corresponding to the negative force. In figure 2, we use a perturbation length of $||\delta\mathbf{r}|| = 0.01$ Å. This consistency with the energy gradient underscores the effectiveness of the PIWSL method, ensuring alignment between predicted total energies and forces, and improving the corresponding RMSE values. As discussed in section 4.2, PIWSL also addresses a key limitation of MLIPs that employ sepa-

rate force branches and do not guarantee the prediction of conservative forces. The proposed method reduces the curl of the predicted forces, as detailed in section D.10, although complete elimination of the curl remains a challenge. In summary, PIWSL minimizes individual energy and force errors, thereby enhancing the overall accuracy of MLIPs.

To further assess PIWSL's impact, we evaluate the robustness during MD simulations of the MLIP models trained with and without PIWSL. We consider MD simulations of the aspirin molecule, with corresponding results presented in figure 3. We measured stability following the approach proposed in Fu et al. (2023). A detailed experimental setup is provided in section D.3.2. The results demonstrate that PIWSL improves the stability of MD simulations for both the direct and gradient-based force prediction models. The simulation times in figure 3 are shorter than those reported by Fu et al. (2023). This difference arises from our choice

to perform MD simulations in the microcanonical (NVE) statistical ensemble instead of the canonical (NVT) statistical ensemble in Fu et al. (2023) to assess stability more accurately without the influence of a thermostat. Results for MD simulations conducted in the canonical statistical ensemble are provided in section D.3.2. Figure 3 shows the time evolution of the total energy during MD simulations, demonstrating that total energy is conserved within a 0.05 % deviation until molecular degradation occurs. These results indicate that PIWSL enhances the stability of MD simulations without compromising total energy conservation.

## 5.4. Fine-Tuning of Foundation Models

In this section, we present a case study demonstrating the application of PIWSL to the fine-tuning of foundation models using a sparse dataset. The detailed experimental setup is provided in section D.4. First, we consider fine-tuning MACE-OFF (large) (Kovács et al., 2023), a MACE model pre-trained on the SPICE dataset (Eastman et al., 2023), including QMugs and liquid water subsets (Isert et al., 2022; Schran et al., 2021).[5] The foundation model is fine-tuned using the aspirin dataset with energies and forces evaluated at the CCSD level of theory (CCSD/cc-pVDZ) (Chmiela et al., 2018). In the following, we consider a more challenging scenario, e.g., where CC energies are extrapolated to the CBS limit, where force labels are typically unavailable due to the high computational cost.

We follow the fine-tuning procedure outlined in the official repository,[6] adjusting the learning rate from $10^{-2}$ to $10^{-3}$ to enhance the final accuracy of the model. The results, presented in Table 3, show that PIWSL significantly improves model accuracy, achieving approximately 40 % lower RMSE values. Notably, the model trained with PIWSL on 512 samples performs similarly to the model trained without it, using nearly doubled training set size, i.e., 950 configurations. These results highlight the data efficiency of our approach. Furthermore, both fine-tuned MACE-OFF models outperform models trained from scratch.

We further evaluate the effectiveness of PIWSL on datasets comprising conformations of a single large molecule. Specifically, we consider the buckyball catcher molecule from the MD22 dataset (Chmiela et al., 2023), which contains 148 atoms. For this experiment, we fine-tune two foundation models, MACE-MP (large) (Batatia et al., 2023) and MACE-OFF (large) (Kovács et al., 2023), using only 50 samples to explore the sparse data regime. As shown in Table 4, PIWSL consistently improves fine-tuning performance. Notably, MACE-OFF outperforms MACE-MP,

---

[5]As the official MACE repository supports only a gradient-based force model, our analysis is restricted to this case.

[6]https://mace-docs.readthedocs.io/en/latest/guide/finetuning.html

---

Table 3: **Results for models trained on the MD17(CCSD) dataset without reference atomic forces.** All models are trained on aspirin data without force labels. Energy RMSE is given in kcal/mol, while force RMSE is in kcal/mol/Å.

| Model | Samples | Epoch | | Baseline | PIWSL |
|---|---|---|---|---|---|
| MACE-OFF | 950 | 100 | E | 1.21 ± 0.00 | **0.72 ± 0.01** |
| | | | F[a] | 6.90 ± 0.01 | **3.77 ± 0.13** |
| MACE-OFF | 512 | 100 | E | 2.03 ± 0.03 | **1.21 ± 0.02** |
| | | | F[a] | 10.96 ± 0.13 | **6.55 ± 0.27** |
| MACE (scratch) | 950 | 1000 | E | 3.27 ± 0.16 | **2.10 ± 0.30** |
| | | | F[a] | 18.05 ± 0.82 | **10.99 ± 0.18** |

[a] We evaluated the accuracy of predicted atomic forces using corresponding force labels provided in the test dataset (CCSD/cc-pVDZ level accuracy).

Table 4: **Energy (E) and force (F) mean-absolute errors (MAEs) for MACE, MACE-OFF, and MACE-MP models fine-tuned on the MD22 dataset.** All models are fine-tuned using 50 training samples of the buckyball-catcher molecule from MD22. Energy MAE is given in kcal/mol, while force MAE is in kcal/mol/Å.

| Model | Train Epoch | | Baseline | PIWSL |
|---|---|---|---|---|
| MACE (scratch) | 800 | E | 1.046 ± 0.095 | **0.745 ± 0.031** |
| | | F | 0.294 ± 0.002 | **0.290 ± 0.002** |
| MACE-MP | 100 | E | **2.111 ± 0.453** | **2.033 ± 0.149** |
| | | F | 0.716 ± 0.009 | **0.700 ± 0.011** |
| MACE-OFF | 100 | E | 1.159 ± 0.154 | **0.992 ± 0.051** |
| | | F | 0.346 ± 0.003 | **0.337 ± 0.001** |

underscoring the importance of selecting an appropriate pre-trained model (Kjeldal & Eriksen, 2024). While a more exhaustive hyperparameter search could potentially yield further gains, such tuning is beyond the scope of this work. Our primary aim is to demonstrate the relative efficacy of the PIWSL approach.

## 5.5. Computational Cost Analysis

**Training time.** Table 5 presents the measured training times for experiments conducted with and without PIWSL. For all models except MACE-OFF, the reported training time corresponds to a single training epoch, averaged over five epochs. These experiments use 1000 training configurations from the ANI-1x dataset with a mini-batch size of six. For MACE-OFF, the training time is measured per epoch on the MD17(CCSD) dataset and averaged over ten epochs, following the setup described in section 5.4.

Table 5 shows that PIWSL increases training time by a factor of approximately two to three relative to the baseline. This increase is primarily due to PIWSL effectively doubling or tripling the number of training labels, which leads to a

Table 5: **Training time comparison for experiments with and without PIWSL.** We measure the time required for a single training epoch and provide the results obtained as an average over five epochs. All training times are provided in seconds.

|  | SchNet | PaiNN | SpinConv | eSCN | Equiformer v2 | MACE-OFF |
|---|---|---|---|---|---|---|
| Baseline | 7.51 | 8.02 | 33.46 | 100.71 | 57.79 | 18.5 |
| PIWSL | 12.84 | 23.48 | 86.28 | 328.48 | 177.55 | 31.7 |

proportional rise in computational cost.[7]

Importantly, the additional training time introduced by PIWSL is negligible compared to the substantial cost and time required for data generation using methods such as DFT or CC, especially as the number of atoms in atomic systems increases. Furthermore, even the large MACE-OFF model requires less than double the training time while achieving an approximately 40 % reduction in error. Finally, it is important to emphasize that PIWSL affects only the training time; the inference time remains unchanged.

**Total number of training epochs.** Table A19 compares extended training iterations, showing that while the baseline model begins to overfit, PIWSL continues to improve performance. Table A20 extends the results from Table 3, further confirming the robustness of PIWSL even under prolonged training of the baseline model. A more detailed discussion is provided in section D.8. This finding is particularly relevant in the sparse data regime, which is the focus of this work.

**Training sample efficiency.** Throughout this section, we have demonstrated that PIWSL enhances model performance. This improvement also implies greater sample efficiency during training. Notably, Table 3 illustrates that PIWSL achieves comparable or superior accuracy while using only half the training data for fine-tuning a foundation model. This property makes PIWSL especially advantageous in data-scarce scenarios, which are central to our investigation.

**Conclusion.** In summary, although PIWSL leads to an increase in training time due to the effective expansion of training samples, the corresponding performance gains justify the additional computational cost—especially when compared to the high cost of generating new labels via DFT or CC. Moreover, PIWSL enables achieving lower errors with a smaller number of training epochs.

---

[7]We can reduce this overhead by employing the 2pt-PISC loss introduced in section D.2.2, which removes the need to estimate the third conformation.

## 6. Discussion and Limitations

This work introduces the PIWSL method, encompassing two distinct physics-informed weakly supervised loss functions, for learning MLIPs. These losses provide the physics-informed weak labels based on the Taylor expansion (PITC loss) and the spatial consistency (PISC loss) of the potential energy. These physics-informed weak labels enable any MLIP to improve its accuracy and robustness, particularly in scenarios characterized by sparse training data, which are common when investigating a new molecular or material system. The improved accuracy and robustness of MLIPs can allow running sufficiently long MD simulations, resulting in a more effective use of active learning approaches. Our extensive experiments demonstrate notable efficacy and efficiency of our method from various aspects: (i) dependence on the training dataset size, (ii) the potential energy prediction variance and robustness in terms of a perturbation on a C–H bond length as well as robustness during MD simulations, and (iii) application to the fine-tuning of foundation models. In particular, it is shown that our PIWSL method enables MLIPs to improve the accuracy of their force predictions even without force labels used explicitly during training. Therefore, PIWSL opens a new possibility for training MLIPs using highly accurate reference methods, such as CCSD(T)/CBS.

**Limitations.** The proposed PIWSL method is tailored to ML models that predict atomic forces and total energies of atomic systems. It cannot be applied to other ML problems unrelated to computational chemistry or materials science. Although this work uses up to the second-order Taylor expansion to obtain weak labels in Eq. (2), employing more sophisticated higher-order ordinary differential equations is a viable alternative. Furthermore, while PIWSL was evaluated on the heterogeneous molecular dataset (ANI-1x), a potential limitation of our analysis is that we performed it only on homogeneous inorganic datasets ($TiO_2$ and LMNTO), which may not fully capture the method's generalizability across diverse inorganic systems.

## Impact Statement

This paper presents work whose goal is to advance the field of machine learning, in particular, machine learning for science. Due to the generic nature of pure science, there are many potential societal consequences of our work in the far future, none of which we feel must be specifically highlighted here.

## Acknowledgements

MN acknowledges support from the Deutsche Forschungsgemeinschaft (DFG, German Research Foundation) under Germany's Excellence Strategy - EXC 2075 – 390740016

and the Stuttgart Center for Simulation Science (SimTech).

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

## A. Related Work

**Addressing Data Sparsity in MLIPs.** Generating training datasets suitable for learning reliable MLIPs is challenging, especially when considering unexplored molecular and material systems. Numerous computationally expensive calculations with either ab initio or first-principles approaches are required for the latter. To mitigate this challenge, active learning (AL) methods, which utilize prediction uncertainty, can be applied (Li et al., 2015; Podryabinkin & Shapeev, 2017; Vandermause et al., 2020; Shuaibi et al., 2021b; Briganti & Lunghi, 2023; Zaverkin et al., 2024b). Furthermore, equivariant MLIPs often reduce required training dataset sizes through improved data efficiency (Batzner et al., 2022; Batatia et al., 2022).

## B. Experimental Setup, Baselines, and Datasets

### B.1. Experimental Setup

**Code for Experiments.** The code used to run our experiments builds upon the recent work (Fu et al., 2023) and extends it to integrate the latest Open Catalyst Project code (Chanussot et al., 2021). We adopt hyper-parameters from the Open Catalyst (OC) project, tuned to the corresponding OC dataset. Note that we do not use this dataset in the presented work, whose main focus is training general-purpose MLIPs that can be used to run molecular dynamics (MD) simulations and geometry optimization. However, the OC dataset has been designed to investigate the latter, making it less suitable for the current study. Our modifications include adjusting the learning rate scheduler, details of which can be found in our repository. For potential energy and force prediction, we utilize mean-absolute error (MAE) and $L_2$-norm (L2MAE) losses with coefficients of 1 and 100, respectively. More details on the model hyperparameters are provided in our repository. For the PITC and PISC loss functions, we use the mean square error (MSE) loss based on an experiment in section D.5. When noise is applied to inorganic material datasets with periodic boundary conditions, atoms outside the unit cell are wrapped back into the cell when noise is applied.

**Training Details.** For training MLIPs, we followed the setup in the Open Catalyst Project. We kept the mini-batch size consistent across all models, as shown in Table A1. We have chosen the mini-batch size based on the maximum memory needed by the most demanding models, such as eSCN and Equiformer v2. All experiments are performed on a single NVIDIA A100 GPU with 81.92 GB memory. To avoid overfitting, we stopped training when the validation loss stopped improving—the specific number of training iterations is provided in Table A2.

We used perturbation vectors $\delta\mathbf{r}$ drawn from a uniform random distribution; see also section 4.4. Particularly, we defined $\delta\mathbf{r} \equiv \epsilon\mathbf{g}$, with each component of $\mathbf{g}$ drawn from a uniform random distribution in the interval $(-1, 1)$. The magnitude $\epsilon$ is also drawn from a uniform random distribution and $\epsilon < \epsilon_{\max}$. This definition of $\delta\mathbf{r}$ differs from the one in Eq. (6), improving the computational efficiency of PIWSL by avoiding the calculation of square root and division.

The remaining hyper-parameters are the coefficients for the PITC and PISC losses ($C_{\mathrm{PITC}}, C_{\mathrm{PISC}}$) and the maximum magnitude $\epsilon_{\max}$ of the perturbation vector $\delta\mathbf{r}$; see Table A3. These hyper-parameters are tuned using Optuna (Akiba et al., 2019) for PaiNN and Equiformer v2. We used 1000 configurations drawn randomly from the original ANI-1x dataset for training. Due to multiple local minima, Optuna identified several optimal hyper-parameter sets in each run. We selected the following representative combinations $(C_{\mathrm{PITC}}, C_{\mathrm{PISC}}, \epsilon_{\max})$ = Case A: (1.2, 0.8, 0.025), Case B: (1.0, 0, 0.01), Case C: (0.1, 0.01, 0.01), Case D: (1.2, 0.01, 0.025), Case E: (1.2, 0.01, 0.01), Case F: (1.2, 0.01, 0.015), Case G: (0.01, 0.001, 0.025), and Case H: (0.1, 0.01, 0.025). We selected the hyper-parameters listed in Table A3 based on the validation dataset performance.

**Splitting Datasets.** We split the original datasets into training, validation, and test sets for our experiments. We shuffled the original datasets using a random seed and selected the training datasets of predefined sizes. For validation, we selected the same number of configurations as in the training dataset if it exceeded 100 configurations; otherwise, we used 100

Table A1: **Employed mini-batch sizes.** We provide mini-batch sized for all datasets and models employed in this work.

|  | ANI-1x | TiO$_2{}^a$ | rMD17 | LMNTO |
|---|---|---|---|---|
| Mini-batch size | 6 | 4 | 16 | 4 |

$^a$ As explained in Table 2, the mini-batch size of the SchNet model was changed to 32 due to high RMSE values observed with a mini-batch size of four as the training dataset size increased. A more detailed discussion of the results for SchNet is provided in section D.1.

Table A2: **Total number of training iterations.** We provide the total number of training iterations for all datasets and training set sizes employed in this work. The number in the parentheses demonstrates the corresponding total number of training epochs.

| $N_{\text{train}}$ | ANI-1x | TiO2 | rMD17 | LMNTO |
|---:|:---:|:---:|:---:|:---:|
| 50 | 7500 (900) | – | – | – |
| 100 | 10,000 (600) | 10,000 (400) | 7500 (1200) | 10,000 (400) |
| 1000 | 40,000 (240) | 10,000 (100) | 10,000 (160) | 10,000 (100) |
| 10,000 | 100,000 (60) | – | – | – |
| 100,000 | 400,000 (30) | – | – | – |
| 1,000,000 | 400,000 (3) | – | – | – |
| 5,000,000 | 420,000 (1) | – | – | – |

Table A3: **Hyper-parameters for the PIWSL loss.** We selected the following hyper-parameter combinations using Optuna (Akiba et al., 2019): $(C_{\text{PITC}}, C_{\text{PISC}}, \epsilon_{\max})$= Case A: (1.2, 0.8, 0.025), Case B: (1.0, 0, 0.01), Case C: (0.1, 0.01, 0.01), Case D: (1.2, 0.01, 0.025), Case E: (1.2, 0.01, 0.01), Case F: (1.2, 0.01, 0.015), Case G: (0.01, 0.001, 0.025), and Case H: (0.1, 0.01, 0.025).

| Dataset | Size | Equiformer v2 | eSCN | PaiNN | SpinConv | SchNet |
|:---|---:|:---:|:---:|:---:|:---:|:---:|
| ANI-1x | 50 | A | C | B | A | A |
| | 100 | A | C | A | D | A |
| | 1000 | D | D | D | B | B |
| | 10,000 | G | C | B | C | C |
| | 100,000 | – | – | C | – | – |
| | 1,000,000 | – | – | C | – | – |
| | 5,000,000 | – | – | B | – | – |
| TiO2 | 100 | A | A | A | A | H |
| | 1000 | G | A | C | A | C |
| LMNTO | 100 | B | B | B | A | B |
| | 1000 | B | A | B | B | B |

configurations to ensure sufficient validation size. For the rMD17 dataset, following (Fu et al., 2023), we used 9000 configurations as a validation dataset and another 10,000 for testing. We used the same test dataset across different sizes of the training datasets for a fair performance comparison. We used 10,000 test configurations for ANI-1x and 1000 for $TiO_2$ and LMNTO.

**Training with Adversarial Directions.** In our experiments, which defined perturbation vectors adversarially (see section E), we determined adversarial directions using Eq. (7). More concretely, we only considered the potential energy, i.e., $\mathbf{y}_{\text{pred}}$ and $\mathbf{y}_{\text{label}}$, to avoid Hessian calculations. In addition, we considered the loss function for the potential energy as $L_{\text{dist}}$ in Eq. (7). The expression of $\mathbf{g} = \nabla_{\mathbf{r}} L_{\text{dist}}$ is then

$$\mathbf{g}_{\mathcal{S}} = \nabla_{\mathbf{r}_i} \sqrt{(E(\mathcal{S}; \boldsymbol{\theta}) - E_{\mathcal{S}}^{\text{ref}})^2} = -\frac{1}{2L_{\text{dist}}}(\mathbf{F}_i(\mathcal{S}; \boldsymbol{\theta}) - \mathbf{F}_{i,\mathcal{S}}^{\text{ref}})(E(\mathcal{S}; \boldsymbol{\theta}) - E_{\mathcal{S}}^{\text{ref}}). \tag{A1}$$

Note that we used the relation $\nabla_{\mathbf{r}_i} E_{\mathcal{S}}^{\text{ref}} = -\mathbf{F}_{i,\mathcal{S}}^{\text{ref}}$ to obtain the final expression. Though $E^{\text{ref}}$ can also be interpreted as a constant regarding atom positions. We have chosen this expression to avoid the case where the adversarial direction $\mathbf{g}$ points to $\mathbf{F}_{i,\mathcal{S}}^{\text{ref}}$ other than the very beginning of the training. Our experiments indicate that the employed expression is slightly better than its alternative. In our experiment, we also randomly flip the sign of $\mathbf{g}_{\mathcal{S}}$ to avoid overfitting to adversarial directions.

### B.2. Baseline Methods

**Data Augmentation with NoisyNode.** In our experiments, we used the NoisyNode approach (Godwin et al., 2022) as one of the baseline methods. This method aims to improve the performance of ML models by adding a perturbation to

node features, i.e., atomic coordinates, and makes ML models recover original labels. This approach enables ML models to be more robust to noise in the data. Although the original method recommends adding a decoder network to learn the denoising process, we do not utilize it following previous work (Liao et al., 2023) and add the perturbation vector to atomic coordinates similar to PIWSL losses, fixing energy and force labels. We implement the NoisyNode approach in our code. Thus, we can expect slightly different behavior compared to the recent work (Godwin et al., 2022; Liao et al., 2023) [8].

**Taylor-Expansion-Based Weak Labels.** Recent work proposed a similar Taylor-expansion-based weak label approach (Cooper et al., 2020). Nonetheless, the loss is different from the one in Eq. (3) as the authors used reference energy and atomic force labels to estimate weak energy labels $E_{\mathcal{S}_{\delta\mathbf{r}}}^{\text{ref}}$ for perturbed atomic configurations $\mathcal{S}_{\delta\mathbf{r}}$

$$E_{\mathcal{S}_{\delta\mathbf{r}}}^{\text{ref}} \approx E_{\mathcal{S}}^{\text{ref}} - \sum_{i=1}^{N_{\text{at}}} \left\langle \delta\mathbf{r}_i, \mathbf{F}_{i,\mathcal{S}}^{\text{ref}} \right\rangle + \mathcal{O}\left(||\delta\mathbf{r}||^2\right). \tag{A2}$$

The trainable parameters of MLIPs are optimized by minimizing the weak label (WL) loss

$$L_{\text{WL}}\left(\mathcal{S}; \boldsymbol{\theta}\right) = \ell\left(E\left(\mathcal{S}_{\delta\mathbf{r}}; \boldsymbol{\theta}\right), E_{\mathcal{S}}^{\text{ref}} - \sum_{i=1}^{N_{\text{at}}} \left\langle \delta\mathbf{r}_i, \mathbf{F}_{i,\mathcal{S}}^{\text{ref}} \right\rangle\right). \tag{A3}$$

Figure 1 (a) illustrates the corresponding approach (Cooper et al., 2020), which computes the energy of a perturbed atomic configuration using a Taylor expansion based on reference energy and atomic force labels. This approach was originally applied to train MLIPs without explicit force labels.

### B.3. Description of the Datasets

**ANI-1x Dataset.** The ANI-1x dataset is a heterogeneous molecular dataset and includes 63,865 organic molecules (with chemical elements H, C, N, and O) whose size ranges from 4 to 64 atoms (Smith et al., 2020). The ML model requires learning total energies and atomic forces for various molecules and their conformations. Total energies and atomic forces are obtained through DFT calculations.

**TiO$_2$ Dataset.** Titanium dioxide (TiO$_2$) is an industrially relevant and well-studied material. TiO$_2$ dataset includes 7815 bulk structures of several TiO$_2$ phases whose reference energies and forces are obtained through DFT calculations (Artrith & Urban, 2016). The number of atoms in a single configuration ranges from 6 to 95.

**rMD17 Dataset.** The rMD17 dataset includes ten small organic molecules, including 100,000 configurations obtained by running MD simulations for each (Christensen & von Lilienfeld, 2020). The ML model requires learning the total energies and atomic forces for each molecule. In this revised version of the MD17 dataset, the molecules are taken from the original MD17 dataset (Chmiela et al., 2017; 2018). However, the energies and forces are recalculated at the PBE/def2-SVP level of theory using very tight SCF convergence and a very dense DFT integration grid.

**LMNTO Sataset.** The Li-Mo-Ni-Ti oxide (LMNTO) is of technological significance as a potential high-capacity positive electrode material for lithium-ion batteries. It exhibits substitutional disorder, with Li, Mo, Ni, and Ti all sharing the same sub-lattice. This dataset includes LMNTO with the composition Li$_8$Mo$_2$Ni$_7$Ti$_7$O$_{32}$ and configurations obtained from an MD simulation, resulting in approximately 2600 structures in total (Cooper et al., 2020).

**MD22 Dataset.** The MD22 dataset (Chmiela et al., 2023) includes seven larger organic molecules, such as a small peptide and a double-walled nanotube, whose size ranges from 42 to 370 atoms. The dataset consists of MD trajectories sampled at temperatures between 400 and 500 K. The ML model requires learning the total energies and atomic forces for each molecule. The energies and forces are calculated at the PBE+MBD level of theory.

---

[8]Note that NoisyNode assumes the unperturbed state to be the equilibrium structure, which may have contributed to the limited performance improvements observed in our experiments. Recently, efforts have been underway to develop an extended version of NoisyNode tailored for non-equilibrium structures, such as (Liao et al., 2024).

Table A4: **Energy (E) and force (F) root-mean-square errors (RMSEs) for the ANI-1x dataset.** The results are obtained by averaging over three independent runs. Energy RMSE is given in kcal/mol, while force RMSE is in kcal/mol/Å.

| | | | $N_{\text{train}} = 50$ | | | $N_{\text{train}} = 10,000$ | |
| | | Baseline | Noisy Nodes | PIWSL | Baseline | Noisy Nodes | PIWSL |
| --- | --- | --- | --- | --- | --- | --- | --- |
| Schnet | E | $90.08 \pm 1.24$ | $\mathbf{76.83 \pm 0.75}$ | $83.90 \pm 2.82$ | $24.88 \pm 0.01$ | $\mathbf{24.86 \pm 0.00}$ | $24.88 \pm 0.00$ |
| | F | $35.49 \pm 0.36$ | $\mathbf{31.13 \pm 0.13}$ | $35.30 \pm 0.87$ | $13.36 \pm 0.01$ | $13.36 \pm 0.00$ | $13.36 \pm 0.00$ |
| PaiNN | E | $212.64 \pm 1.14$ | $440.11 \pm 11.68$ | $\mathbf{121.36 \pm 4.13}$ | $19.14 \pm 0.38$ | $165.25 \pm 4.87$ | $\mathbf{14.10 \pm 0.14}$ |
| | F | $22.61 \pm 0.04$ | $22.50 \pm 0.22$ | $\mathbf{20.83 \pm 0.28}$ | $8.24 \pm 0.10$ | $9.22 \pm 0.09$ | $\mathbf{7.89 \pm 0.02}$ |
| SpinConv | E | $222.75 \pm 7.12$ | $219.85 \pm 6.99$ | $\mathbf{175.38 \pm 9.77}$ | $19.42 \pm 0.67$ | $46.31 \pm 10.31$ | $\mathbf{18.81 \pm 0.60}$ |
| | F | $\mathbf{24.88 \pm 0.88}$ | $24.61 \pm 0.35$ | $25.12 \pm 0.58$ | $10.31 \pm 0.33$ | $10.78 \pm 0.66$ | $\mathbf{9.94 \pm 0.12}$ |
| eSCN | E | $517.17 \pm 31.98$ | $583.90 \pm 33.04$ | $\mathbf{454.40 \pm 11.10}$ | $12.65 \pm 0.63$ | $165.30 \pm 33.11$ | $\mathbf{10.66 \pm 0.31}$ |
| | F | $22.51 \pm 0.09$ | $24.04 \pm 0.15$ | $\mathbf{22.28 \pm 0.08}$ | $5.11 \pm 0.30$ | $11.51 \pm 0.23$ | $\mathbf{4.35 \pm 0.15}$ |
| Equiformer | E | $498.58 \pm 17.44$ | $630.32 \pm 0.32$ | $\mathbf{433.88 \pm 79.63}$ | $8.03 \pm 0.21$ | $970.95 \pm 236.90$ | $\mathbf{7.77 \pm 0.14}$ |
| | F | $22.86 \pm 0.04$ | $22.92 \pm 0.00$ | $\mathbf{22.72 \pm 0.04}$ | $\mathbf{2.97 \pm 0.00}$ | $29.28 \pm 5.63$ | $2.98 \pm 0.00$ |

## C. Differences in Gradients for Physics-Informed Losses

The following considers the gradients of the proposed two losses. First, considering squared errors, we obtain the following gradients of the loss in Eq. (A2) with respect to trainable parameters

$$\frac{\mathrm{d}\mathcal{L}_{\text{WL}}}{\mathrm{d}\boldsymbol{\theta}} = 2\left( E\left(\mathcal{S}_{\delta\mathbf{r}}; \boldsymbol{\theta}\right) - E_{\mathcal{S}}^{\text{ref}} + \sum_{i=1}^{N_{\text{at}}} \langle \delta\mathbf{r}_i, \mathbf{F}_{i,\mathcal{S}}^{\text{ref}} \rangle \right) \frac{\mathrm{d}E\left(\mathcal{S}_{\delta\mathbf{r}}; \boldsymbol{\theta}\right)}{\mathrm{d}\boldsymbol{\theta}}. \tag{A4}$$

In contrast, for the PITC loss in Eq. (3) we obtain

$$\frac{\mathrm{d}\mathcal{L}_{\text{PITC}}}{\mathrm{d}\boldsymbol{\theta}} = 2\left( E\left(\mathcal{S}_{\delta\mathbf{r}}; \boldsymbol{\theta}\right) - E\left(\mathcal{S}; \boldsymbol{\theta}\right) + \sum_{i=1}^{N_{\text{at}}} \langle \delta\mathbf{r}_i, \mathbf{F}_i\left(\mathcal{S}; \boldsymbol{\theta}\right) \rangle \right) \times$$
$$\left( \frac{\mathrm{d}E\left(\mathcal{S}_{\delta\mathbf{r}}; \boldsymbol{\theta}\right)}{\mathrm{d}\boldsymbol{\theta}} - \frac{\mathrm{d}E\left(\mathcal{S}; \boldsymbol{\theta}\right)}{\mathrm{d}\boldsymbol{\theta}} + \sum_{i=1}^{N_{\text{at}}} \frac{\mathrm{d}\langle \delta\mathbf{r}_i, \mathbf{F}_i\left(\mathcal{S}; \boldsymbol{\theta}\right) \rangle}{\mathrm{d}\boldsymbol{\theta}} \right). \tag{A5}$$

The above equations indicate that the direction of the derivative of the PITC loss in Eq. (A5) is different from that of the weak label loss because of the incorporation of the predicted potential energy at the original and the force at the reference point. The gradient of PISC loss in Eq. (4) reads

$$\frac{\mathrm{d}\mathcal{L}_{\text{PISC}}}{\mathrm{d}\boldsymbol{\theta}} = 2\left( E\left(\mathcal{S}; \boldsymbol{\theta}\right) - \sum_{i=1}^{N_{\text{at}}} \langle \delta\mathbf{r}_i, \mathbf{F}_i\left(\mathcal{S}; \boldsymbol{\theta}\right) \rangle - E\left(\mathcal{S}_{\delta\mathbf{r}'}; \boldsymbol{\theta}\right) + \sum_{i=1}^{N_{\text{at}}} \langle \delta\mathbf{r}_i'', \mathbf{F}_i\left(\mathcal{S}_{\delta\mathbf{r}'}; \boldsymbol{\theta}\right) \rangle \right)$$
$$\times \left( \frac{\mathrm{d}E\left(\mathcal{S}; \boldsymbol{\theta}\right)}{\mathrm{d}\boldsymbol{\theta}} - \sum_{i=1}^{N_{\text{at}}} \frac{\mathrm{d}\langle \delta\mathbf{r}_i, \mathbf{F}_i\left(\mathcal{S}; \boldsymbol{\theta}\right) \rangle}{\mathrm{d}\boldsymbol{\theta}} - \frac{\mathrm{d}E\left(\mathcal{S}_{\delta\mathbf{r}'}; \boldsymbol{\theta}\right)}{\mathrm{d}\boldsymbol{\theta}} + \sum_{i=1}^{N_{\text{at}}} \frac{\mathrm{d}\langle \delta\mathbf{r}_i'', \mathbf{F}_i\left(\mathcal{S}_{\delta\mathbf{r}'}; \boldsymbol{\theta}\right) \rangle}{\mathrm{d}\boldsymbol{\theta}} \right). \tag{A6}$$

## D. Experiments

### D.1. Benchmark Results

The following section provides additional results, complementing those provided in the main text.

**Additional Results for ANI-1x.** Table A4 provides results for ANI-1x dataset and a training set sizes of 50 or 10,000; see also figure 2. The table demonstrates a considerable reduction of energy and force RMSEs for models trained using small training dataset sizes of 50 configurations. Furthermore, we find around 5 to 25 % error reduction for a larger training set size of 10,000, indicating the effectiveness of the PIWSL method for relatively large training set sizes. Finally, we provide

Table A5: **Energy and force erorrs of PaiNN model trained on the ANI-1x dataset with 100,000, 1,000,000, and 5,000,000 samples.** The results are obtained by averaging over three independent runs. Energy errors are given in kcal/mol, while force errors are in kcal/mol/Å.

| | $N_{\text{train}}$ | Force MAE | Force RMSE | Energy MAE | Energy RMSE |
|---|---|---|---|---|---|
| Baseline | 100,000 | **0.92 ± 0.00** | **3.70 ± 0.01** | 4.28 ± 0.15 | 6.14 ± 0.21 |
| PIWSL | | **0.91 ± 0.01** | **3.72 ± 0.04** | **4.07 ± 0.17** | **5.83 ± 0.20** |
| Baseline | 1,000,000 | **0.67 ± 0.00** | **2.74 ± 0.01** | 4.90 ± 0.23 | 6.56 ± 0.26 |
| PIWSL | | 0.68 ± 0.00 | **2.77 ± 0.04** | **4.48 ± 0.05** | **6.06 ± 0.01** |
| Baseline | 5,000,000[a] | **0.53** | **2.18** | 3.94 | 5.24 |
| PIWSL | | 0.55 | 2.24 | **3.25** | **4.75** |

[a] Because of the computational cost, we performed only one training in the case of 5,000,000 training samples.

Table A6: **Energy (F) and force (F) root-mean-square errors (RMSEs) for the TiO2 dataset obtained for the SchNet model with a mini-batch size of four.** The results are obtained by averaging over three independent runs. Energy RMSE is given in kcal/mol, while force RMSE is in kcal/mol/Å.

| | | $N_{\text{train}} = 100$ | | | $N_{\text{train}} = 1000$ | | |
|---|---|---|---|---|---|---|---|
| | | Baseline | Noisy Nodes | PIWSL | Baseline | Noisy Nodes | PIWSL |
| SchNet | E | 18.85 ± 0.00 | **17.48 ± 0.00** | 17.58 ± 0.00 | 35.58 ± 0.00 | 58.08 ± 18.44 | **15.28 ± 0.12** |
| | F | 2.74 ± 0.00 | **2.51 ± 0.00** | 2.74 ± 0.00 | 6.54 ± 0.00 | 18.40 ± 0.00 | **3.61 ± 0.27** |

a result of PaiNN model trained on ANI-1x dataset with 100,000, 1,000,000, and 5,000,000 samples in Table A5, which demonstrates that PIWSL still improves the performance around 5% to 10%of the energy RMSE.

**Additional Results for SchNet Applied to TiO$_2$.** In Table 2, we set the mini-batch size to 32 for training the SchNet model. This adjustment was made because training SchNet with a small mini-batch size of four increases RMSE values with a growing training dataset size. Table A6 demonstrates the performance of the SchNet model for a mini-batch size of four. Table A7 provides the results obtained for the SchNet model with a mini-batch size of four for the following training set sizes: 100, 200, 500, and 1000. This figure demonstrates that SchNet, with a mini-batch size of four, reaches its best performed with $N_{\text{train}} = 200$. These results indicate the difficulty of learning training data statistics from small mini-batches, probably due to the limited expressive power of SchNet.

**Results for LMNTO.** Table A8 presents RMSE errors for LMNTO (Cooper et al., 2020). PIWSL shows the error reduction for most cases for this benchmark dataset, especially for small training set sizes (i.e., a training set of 100 configurations).

**Molecular Dynamics Trajectories for Large Molecules – MD22.** Table A9 evaluates the impact of PIWSL on datasets containing conformations of a single large molecule. For this purpose, we selected the buckyball catcher molecule with $n_{\text{atom}} = 148$ atoms. To demonstrate the applicability of PIWSL, we trained a MACE model (Batatia et al., 2022) to prove that PIWSL enhances even the performance of a recent state-of-the-art model. The model structure and training configuration were based on those provided in the official repository[9] with slight modifications: "max_L" was set to one and mini-batch size was adjusted to four. Following the original training setup (Chmiela et al., 2023), we randomly sampled 600 configurations for the training dataset and 400 for the validation dataset and retained the remaining 5102 configurations for testing. To further validate PIWSL's effectiveness in sparse data scenarios, we prepared a smaller training dataset comprising only 50 configurations while keeping the validation dataset size unchanged. The model was trained for 450 and 800 epochs for the 600-sample and 50-sample training datasets, respectively. The results presented in Table A9 demonstrate that our approach remains effective on average, particularly in the sparse data regime. In this study, the coefficient of the PITC and PISC losses are set as 0.01 and 0.001 with $\epsilon_{\text{max}} = 0.01$.

[9] https://mace-docs.readthedocs.io/en/latest/examples/training_examples.html

Table A7: **Training dataset size dependence of SchNet with a mini-batch size of four.** The results are presented for the $TiO_2$ dataset and are obtained by averaging over three independent runs. Energy RMSE is given in kcal/mol, while force RMSE is in kcal/mol/Å.

|  |  | $N_{\text{train}} = 100$ | $N_{\text{train}} = 200$ | $N_{\text{train}} = 500$ | $N_{\text{train}} = 1000$ |
|---|---|---|---|---|---|
| SchNet | E | $18.85 \pm 0.00$ | $\mathbf{16.28 \pm 0.00}$ | $24.42 \pm 0.00$ | $35.58 \pm 0.00$ |
|  | F | $2.74 \pm 0.00$ | $\mathbf{2.56 \pm 0.00}$ | $4.43 \pm 0.00$ | $6.54 \pm 0.00$ |

Table A8: **Energy (E) and force (F) root-mean-square errors (RMSEs) for the LMNTO dataset.** The results are obtained by averaging over three independent runs. Energy RMSE is given in kcal/mol, while force RMSE is in kcal/mol/Å.

| Model |  | $N_{\text{train}} = 100$ | | | $N_{\text{train}} = 1000$ | | |
|---|---|---|---|---|---|---|---|
|  |  | Baseline | NoisyNode | PIWSL | Baseline | NoisyNode | PIWSL |
| SchNet | E | $4.46 \pm 0.00$ | $6.10 \pm 0.00$ | $\mathbf{4.45 \pm 0.00}$ | $\mathbf{3.09 \pm 0.00}$ | $3.25 \pm 0.00$ | $\mathbf{3.09 \pm 0.00}$ |
|  | F | $9.24 \pm 0.00$ | $\mathbf{8.31 \pm 0.00}$ | $9.24 \pm 0.00$ | $\mathbf{5.09 \pm 0.00}$ | $5.21 \pm 0.00$ | $\mathbf{5.09 \pm 0.00}$ |
| PaiNN | E | $6.91 \pm 0.02$ | $7.09 \pm 0.04$ | $\mathbf{5.99 \pm 0.02}$ | $3.26 \pm 0.01$ | $4.61 \pm 0.03$ | $\mathbf{2.98 \pm 0.01}$ |
|  | F | $\mathbf{4.75 \pm 0.00}$ | $7.20 \pm 0.01$ | $\mathbf{4.75 \pm 0.00}$ | $\mathbf{2.03 \pm 0.00}$ | $2.55 \pm 0.00$ | $\mathbf{2.03 \pm 0.00}$ |
| SpinConv | E | $7.90 \pm 0.00$ | $7.83 \pm 0.04$ | $\mathbf{7.83 \pm 0.01}$ | $4.90 \pm 0.33$ | $7.20 \pm 0.06$ | $\mathbf{3.95 \pm 0.02}$ |
|  | F | $\mathbf{4.63 \pm 0.01}$ | $5.14 \pm 0.04$ | $4.71 \pm 0.02$ | $1.81 \pm 0.01$ | $2.33 \pm 0.00$ | $\mathbf{1.74 \pm 0.00}$ |
| eSCN | E | $7.92 \pm 0.00$ | $7.92 \pm 0.00$ | $7.92 \pm 0.00$ | $7.93 \pm 0.00$ | $7.93 \pm 0.00$ | $\mathbf{6.40 \pm 0.14}$ |
|  | F | $4.67 \pm 0.01$ | $7.59 \pm 0.02$ | $\mathbf{4.64 \pm 0.01}$ | $1.54 \pm 0.00$ | $1.98 \pm 0.06$ | $\mathbf{1.53 \pm 0.00}$ |
| Equiformer v2 | E | $7.40 \pm 0.03$ | $7.92 \pm 0.00$ | $\mathbf{7.32 \pm 0.08}$ | $\mathbf{3.57 \pm 0.05}$ | $7.04 \pm 0.03$ | $3.60 \pm 0.02$ |
|  | F | $4.26 \pm 0.00$ | $7.60 \pm 0.02$ | $\mathbf{4.24 \pm 0.02}$ | $\mathbf{1.34 \pm 0.00}$ | $1.99 \pm 0.00$ | $\mathbf{1.34 \pm 0.00}$ |

In addition to the experiment of training from scratch, we conducted an additional study using the MACE foundation model to evaluate whether PIWSL could be effectively applied in the context of foundation models with finetuning. Specifically, we utilized two foundation models: MACE-MP ("large") (Batatia et al., 2023), a universal model, and MACE-OFF ("large_off") (Kovács et al., 2023), a model designed for organic force fields. Training was performed on the buckyball catcher molecule from the MD22 dataset. To simulate finetuning on a smaller dataset, the models were trained using 50 samples, consistent with the previous experiment. The official settings from the respective repositories were used, with adjustments made to the mini-batch size (set to 4) and the number of epochs (set to 100).

The results, presented in Table 4, demonstrate that PIWSL is effective for the fine-tuning of foundation models. Moreover, the result indicates that the MACE-MP model provides worse results than MACE-OFF, emphasizing the importance of selecting an appropriate pre-trained model. For this experiment, we employed the second-order PITC loss and the two-point consistency loss defined in Eq. (A14) and Eq. (A11), with the coefficients set to 0.08.

Table A9: **Energy (E) and force (F) mean-absolute errors (MAEs) for the MD22 dataset.** We have chosen buckyball catcher for our experiments. The results are obtained by averaging over three independent runs. Energy MAE is given in kcal/mol/atom, while force MAE is in kcal/mol/Å.

| Dataset | Model |  | $N_{\text{train}} = 50$ | | $N_{\text{train}} = 600$ | |
|---|---|---|---|---|---|---|
|  |  |  | Baseline | PIWSL | Baseline | PIWSL |
| Buckyball catcher | MACE | E | $1.046 \pm 0.095$ | $\mathbf{0.745 \pm 0.031}$ | $0.587 \pm 0.107$ | $\mathbf{0.548 \pm 0.015}$ |
|  |  | F | $0.294 \pm 0.002$ | $\mathbf{0.290 \pm 0.002}$ | $0.082 \pm 0.001$ | $0.082 \pm 0.001$ |

## D.2. Different Configurations for the Physics-Informed Spatial-Consistency Loss

### D.2.1. TRIANGLE-BASED

Table A10: **Results for different configurations of the PISC loss.** The presented numerical values are the root mean square errors (RMSEs) for the ANI-1x dataset (Smith et al., 2020). Energy (in kcal/mol) and force (in kcal/mol/Å) errors are obtained by averaging over three independent runs. All models are trained using 1000 configurations. The case 1, 2, and 3 correspond to Eq. (4), Eq. (A8) and Eq. (A9), respectively.

| Model | | Baseline | PISC (Case 1) | PISC (Case 2) | PISC (Case 3) |
|---|---|---|---|---|---|
| PaiNN | E | 60.11 | **45.24** | 46.32 | 57.29 |
| | F | 13.10 | **12.33** | 12.42 | 13.28 |

In section 4.2, we consider the following form of the PISC loss

$$L_{\mathrm{PISC}}\left(\mathcal{S};\boldsymbol{\theta}\right) = \ell\left(E\left(\mathcal{S}_{\delta\mathbf{r}};\boldsymbol{\theta}\right), E\left(\mathcal{S}_{\delta\mathbf{r}'};\boldsymbol{\theta}\right) - \sum_{i=1}^{N_{\mathrm{at}}}\langle\delta\mathbf{r}_i'', \mathbf{F}_i\left(\mathcal{S}_{\delta\mathbf{r}'};\boldsymbol{\theta}\right)\rangle\right), \tag{A7}$$

where $\delta\mathbf{r}, \delta\mathbf{r}', \delta\mathbf{r}''$ are related as $\delta\mathbf{r}' + \delta\mathbf{r}'' = \delta\mathbf{r}$. In this section, as a variant of Eq. (4), we also consider the following three PISC losses

$$L_{\mathrm{PISC,Case\,2}}\left(\mathcal{S};\boldsymbol{\theta}\right) = \ell\left(E\left(\mathcal{S};\boldsymbol{\theta}\right) - \sum_{i=1}^{N_{\mathrm{at}}}\langle\delta\mathbf{r}_i, \mathbf{F}_i\left(\mathcal{S};\boldsymbol{\theta}\right)\rangle, E\left(\mathcal{S}_{\delta\mathbf{r}'};\boldsymbol{\theta}\right) - \sum_{i=1}^{N_{\mathrm{at}}}\langle\delta\mathbf{r}_i'', \mathbf{F}_i\left(\mathcal{S}_{\delta\mathbf{r}'};\boldsymbol{\theta}\right)\rangle\right), \tag{A8}$$

$$L_{\mathrm{PISC,Case\,3}}\left(\mathcal{S};\boldsymbol{\theta}\right) = \ell\left(E\left(\mathcal{S}_{\delta\mathbf{r}'};\boldsymbol{\theta}\right), E\left(\mathcal{S}_{\delta\mathbf{r}};\boldsymbol{\theta}\right) - \sum_{i=1}^{N_{\mathrm{at}}}\langle-\delta\mathbf{r}_i'', \mathbf{F}_i\left(\mathcal{S}_{\delta\mathbf{r}};\boldsymbol{\theta}\right)\rangle\right), \tag{A9}$$

where the point at $\mathbf{r} + \delta\mathbf{r}$ is the point where PIRC loss is imposed (see Eq. (3)). The results are provided in Table A10 and indicate that Eq. (4) (Case 1) shows a better performance than the other cases for both the potential energy and the force predictions. In this study, we used the ANI-1x dataset with 1000 training samples different from the one used to train the model used in the main body to avoid overfitting on the test dataset. For the coefficient of the PITC and PISC losses, we used 0.1 and 0.001 with $\epsilon_{\mathrm{max}} = 0.01$.

### D.2.2. FURTHER VARIATIONS

**Two-Point Spacial Consistency.** The flexibility of the PISC loss allows us to explore additional forms of spatial consistency. For example, instead of using a triangular configuration, we can impose spatial consistency between two points at $\mathbf{r}$ and $\mathbf{r} + \delta\mathbf{r}$, leading to the following expression:

$$L_{\mathrm{PISC,2pt}}\left(\mathcal{S};\boldsymbol{\theta}\right) = \ell\left(E\left(\mathcal{S};\boldsymbol{\theta}\right), E\left(\mathcal{S}_{\delta\mathbf{r}};\boldsymbol{\theta}\right) - \sum_{i=1}^{N_{\mathrm{at}}}\langle-\delta\mathbf{r}_i, \mathbf{F}_i\left(\mathcal{S}_{\delta\mathbf{r}};\boldsymbol{\theta}\right)\rangle\right). \tag{A10}$$

While not thoroughly investigated, we empirically observed that this loss delivers competitive performance when applied with the same coefficient value as the PITC loss.

**Two-Point Spacial Consistency with Label.** Since the training sample includes a label, we can utilize this label information instead of predicting the potential energy of the original conformation to enforce two-point spatial consistency.

$$L_{\mathrm{PISC,2ptwl}}\left(\mathcal{S};\boldsymbol{\theta}\right) = \ell\left(E_{\mathcal{S}}^{\mathrm{ref}}, E\left(\mathcal{S}_{\delta\mathbf{r}};\boldsymbol{\theta}\right) - \sum_{i=1}^{N_{\mathrm{at}}}\langle-\delta\mathbf{r}_i, \mathbf{F}_i\left(\mathcal{S}_{\delta\mathbf{r}};\boldsymbol{\theta}\right)\rangle\right). \tag{A11}$$

We observed that this loss function performs well when applying the MACE model to datasets without force labels, as discussed in subsection 5.4.

**Second-order Term Consideration for PITC Loss.** In Eq. (2), we considered only the first-order Taylor expansion. Here, we introduce a straightforward approach to approximately account for the second-order term. Using the energy and forces at $\mathbf{r} + \delta\mathbf{r}$, the second-order derivative of the potential energy can be approximated using the explicit finite difference method as:

$$\delta\mathbf{r}^\mu \delta\mathbf{r}^\nu \partial_\mu \partial_\nu E = \delta\mathbf{r}^\mu \delta\mathbf{r}^\nu \partial_\mu F_\nu \simeq \langle \delta\mathbf{r}, \mathbf{F}\left(\mathcal{S}_{\delta\mathbf{r}}; \boldsymbol{\theta}\right) - \mathbf{F}\left(\mathcal{S}; \boldsymbol{\theta}\right) \rangle, \tag{A12}$$

Then, Eq. (2) with the above second-order term can be written as:

$$E\left(\mathcal{S}_{\delta\mathbf{r}}; \boldsymbol{\theta}\right) \approx E\left(\mathcal{S}; \boldsymbol{\theta}\right)$$
$$- \sum_{i=1}^{N_{\mathrm{at}}} \left[ \langle \delta\mathbf{r}_i, \mathbf{F}_i\left(\mathcal{S}; \boldsymbol{\theta}\right) \rangle + k_{\mathrm{2nd}} \langle \delta\mathbf{r}_i, \mathbf{F}_i\left(\mathcal{S}_{\delta\mathbf{r}}; \boldsymbol{\theta}\right) - \mathbf{F}_i\left(\mathcal{S}; \boldsymbol{\theta}\right) \rangle \right] + \mathcal{O}\left(\|\delta\mathbf{r}\|^3\right),$$
$$= E\left(\mathcal{S}; \boldsymbol{\theta}\right) - \sum_{i=1}^{N_{\mathrm{at}}} \langle \delta\mathbf{r}_i, (1 - k_{\mathrm{2nd}})\mathbf{F}_i\left(\mathcal{S}; \boldsymbol{\theta}\right) + k_{\mathrm{2nd}}\mathbf{F}_i\left(\mathcal{S}_{\delta\mathbf{r}}; \boldsymbol{\theta}\right) \rangle + \mathcal{O}\left(\|\delta\mathbf{r}\|^3\right), \tag{A13}$$

where $k_{\mathrm{2nd}}$ is a numeric parameter that controls the contribution of the second-order term. Note that setting $k_{\mathrm{2nd}} = 0.5$ recovers the exact second-order expression, while $k_{\mathrm{2nd}} = 0$ reverts to the first-order approximation.

Using Eq. (A13), the expression of the second-order PITC loss is given as:

$$L_{\mathrm{PITC}}\left(\mathcal{S}; \boldsymbol{\theta}\right) = \ell\Big( E\left(\mathcal{S}_{\delta\mathbf{r}}; \boldsymbol{\theta}\right), E\left(\mathcal{S}; \boldsymbol{\theta}\right) - \sum_{i=1}^{N_{\mathrm{at}}} \langle \delta\mathbf{r}_i, (1 - k_{\mathrm{2nd}})\mathbf{F}_i\left(\mathcal{S}; \boldsymbol{\theta}\right) + k_{\mathrm{2nd}}\mathbf{F}_i\left(\mathcal{S}_{\delta\mathbf{r}}; \boldsymbol{\theta}\right) \rangle \Big), \tag{A14}$$

Note that the second-order term becomes significant under the following conditions: (1) the accuracy of the MLIP model surpasses the contribution of the first-order term, and (2) the original conformation is near the equilibrium state. The latter can be understood as follows. Considering a spring model as a two-body interaction, the potential energy can be expressed as:

$$V(\mathbf{r}) = \frac{k_{\mathrm{bond}}}{2}(\mathbf{r} - \mathbf{r}_0)^2, \tag{A15}$$

where $k_{\mathrm{bond}}$ is the constant characterizing the strength of the two-body interaction, and $\mathrm{r}_0$ denotes the equilibrium bond length. Introducing a small perturbation $\delta\mathbf{r}$ in the bond length, the potential energy becomes:

$$V(\mathbf{r} + \delta\mathbf{r}) = \frac{k_{\mathrm{bond}}}{2}(\mathbf{r} - \mathbf{r}_0 + \delta\mathbf{r})^2 = V(\mathbf{r}) + 2k_{\mathrm{bond}}\langle \mathbf{r} - \mathbf{r}_0, \delta\mathbf{r} \rangle + \frac{k_{\mathrm{bond}}}{2}\delta\mathbf{r}^2. \tag{A16}$$

At the equilibrium state ($\mathbf{r} = \mathbf{r}_0$), the above equation demonstrates that the second-order term becomes dominant. Consequently, the second-order accuracy of the PITC loss function becomes crucial in such scenarios.

**PITS for Curl of Forces** Another potential direction is enforcing a reduction in the curl of the forces. This can be achieved by leveraging Stokes' theorem: $\int_\Sigma \nabla \times \mathbf{F} \cdot d\mathbf{S} = \oint_{\partial\Sigma} \mathbf{F} \cdot d\mathbf{l} = 0$ where $\Sigma$ represents a specific surface regime, and $\partial\Sigma$ denotes its boundary. Similar to the PISC loss, the right-hand side of this equation can be effectively described by considering a triangular configuration, where the midpoints of the three sides correspond to $\mathbf{r}, \mathbf{r} + \delta\mathbf{r}, \mathbf{r} + \delta\mathbf{r}'$.

### D.3. Detailed Setups for Qualitative Analysis

#### D.3.1. C–H POTENTIAL ENERGY PROFILE OF ASPIRIN

This section describes the detailed setup and procedure for section 5.3. First, we trained PaiNN with and without PIWSL losses using the aspirin data from rMD17 with training set sizes of 100 and 200. For PIWSL, we used $(C_{\mathrm{PITC}}, C_{\mathrm{PISC}}, \epsilon_{\mathrm{max}}) = (1.2, 0.01, 0.015)$. The other experimental setups are the same as for rMD17 experiments presented in section B.1. We used the PaiNN model with gradient-based forces to obtain the reference model and tuned the model hyper-parameter with Optuna (Akiba et al., 2019). The obtained models' performance is provided in Table A11. Then, we prepared the aspirin molecule structures, including the corresponding atomic coordinates and atomic types. For these structures, we perturbed one of the C-H bonds with a bond length from 0.8 Å to 1.8 Å. We prepared 100 structures and estimated the corresponding potential energy with the pre-trained models. The aspirin data is provided in our publicly available source code.

Table A11: **Performance of PaiNN employed in figure 2 (c, d).** All the models other than the reference model ($N_{\text{train}} = 1000$) use the OC20's hyper-parameters. For the reference model, we tuned the hyper-parameters of PaiNN model following the original paper (Schütt et al., 2021).

|  |  | $N_{\text{train}} = 100$ | | $N_{\text{train}} = 200$ | | $N_{\text{train}} = 1000$ |
|---|---|---|---|---|---|---|
|  |  | Baseline | PIWSL | Baseline | PIWSL | Baseline |
| PaiNN | E | 6.55 | **5.64** | 5.11 | **4.48** | 0.68 |
|  | F | 7.38 | 7.36 | 3.95 | 3.97 | 1.44 |

### D.3.2. MD Simulation Stability Analysis

**NVE-Ensembles** This section describes the detailed setup and the procedure for our analysis of MD simulations in section 5.3. Because our implementation builds upon the source code provided by Fu et al. (2023), we used their scripts for performing MD simulations. However, we added a minor modification to enable MD simulations in the microcanonical (NVE) statistical ensemble, i.e., the particle position and velocity are updated with velocity Verlet algorithm(Verlet, 1967) [10]. We set the initial temperature to 300 K and the integration time step to 0.5 fs for all simulations. As defined by Fu et al. (2023), the stability of an MD simulation for a target molecule is defined as the time $T$ during which the bond lengths satisfy the following condition

$$\max_{(i,j)\in\mathcal{B}} |(||\mathbf{x}_i(T) - \mathbf{x}_j(T)||) - b_{i,j}| > \Delta, \tag{A17}$$

where $\mathcal{B}$ denotes the set of all bonds, $\{i, j\}$ denote the pair of bonded atoms, and $b_{i,j}$ denotes the equilibrium bond length. Following Fu et al. (2023), we set $\Delta = 0.5$Å. This definition indicates when the molecule experiences significant structural changes during the MD simulation.

We trained PaiNN and Equiformer v2 models with and without PIWSL losses using the aspirin data from rMD17. We used training set sizes of 100 and 200 for PaiNN-GF and 1000 for PaiNN and Equiformer v2 with direct force. The corresponding stability values are presented in Table A12. The hyperparameters for the PIWSL loss are provided in Table A13. To train direct force model with 1000 samples, we used second-order PITC and PISC losses because of their good accuracy. The results for the stability of the PaiNN (direct and gradient-based force) and Equiformer v2 models are shown in Table A14. To select the hyperparameters in Table A13, a series of MD simulations with a fixed random number seed for the initial atomic velocities was used. For our final stability results in Table A12, a series of MD simulations with three different random seeds for velocities and the selected hyperparameters was used.

**NVT-Ensembles** To investigate the effect of thermostats, we also performed MD simulations in the canonical (NVT) statistical ensemble, where temperature is maintained constant. To keep the temperature constant, we used Nosé-Hoover thermostat (Nosé, 1984; Hoover, 1985). The initial and target temperatures were both set to 300 K for all simulations. The integration time step was set to 0.5 fs and the characteristic parameter $\tau$ for the thermostat is set to 20 fs. The result, shown in figure A1, demonstrate that the thermostat stabilizes the simulations by mitigating the increase in kinetic energy.

### D.4. Training setup for MD17-CCSD(T) Experiments

In this section, we provide the training setup of finetuning of MACE-OFF discussed in section 5.4. The considered model is MACE-OFF (large) (Kovács et al., 2023) which is the MACE model pretrained on SPICE (Eastman et al., 2023), QMugs (Isert et al., 2022), and liquid water (Schran et al., 2021) datasets. The foundation model is finetuned on the aspirin molecule data in CCSD dataset (Chmiela et al., 2018) whose potential energy is obtained at quantum-chemical CCSD level of accuracy. The data includes 1500 samples which are split into 950/50/500 as train/validation/test datasets. To emulate general coupling-cluster method dataset, we only use potential energy label for training. We utilize the set up provided in the official repository[11] with modifying learning rate from $10^{-2}$ to $10^{-3}$, to improve the performance. For the model trained scratch, we use the same architecture with setting energy weight to 40 and learning rate to $10^{-3}$ to improve the performance. For PIWSL, we used the second-order PITC and 2pt-with-label PISC losses Equation A10 with the following parameters:

---

[10]Note that the total energy conservation necessary for the microcanonical statistical ensemble is in general not perfectly satisfied due to the numerical error, in particular, when the force is not calculated as the curl of the force.

[11]https://mace-docs.readthedocs.io/en/latest/guide/finetuning.html

Table A12: **Stability of the models employed in the MD analysis.** The presented numerical values are the stability defined by Eq. (A17) measured in ps. The results are obtained as an average over three different random seeds for the initial velocity of the atoms in the target aspirin molecule. "GF" denotes the gradient-based force prediction.

| | | $N_{\text{train}} = 100$ | $N_{\text{train}} = 200$ | $N_{\text{train}} = 1000$ |
|---|---|---|---|---|
| PaiNN | Baseline | – | – | $2.68 \pm 0.13$ |
| | PIWSL | – | – | $\mathbf{4.65 \pm 0.72}$ |
| Equiformer | Baseline | – | – | $14.53 \pm 8.61$ |
| | PIWSL | – | – | $\mathbf{24.65 \pm 8.24}$ |
| PaiNN-GF | Baseline | $3.25 \pm 3.98$ | $220.5 \pm 137.7$ | – |
| | PIWSL | $\mathbf{15.07 \pm 10.09}$ | $\mathbf{267.7 \pm 56.0}$ | – |

Table A13: **Hyper-parameters for the PIWSL loss used in the MD analysis.** We used the following hyper-parameter for MD simulation analysis: $(C_{\text{PITC}}, C_{\text{PISC}}, \epsilon_{\text{max}})=$ Case $\alpha$: (0.01, 0.001, 0.025), Case $\beta$: (1.2, 0.01, 0.01), Case $\gamma$: (1.2, 0.01, 0.025), Case $\delta$: (1.2, 0.01, 0.015), Case $\epsilon$: (0.1, 0.01, 0.01), and Case $\zeta$: (1.0, 0., 0.01). "GF" denotes the gradient-based force prediction.

| Dataset | Size | Equiformer v2 | PaiNN | PaiNN-GF |
|---|---|---|---|---|
| rMD17 | 100 | $\alpha$ | $\delta$ | $\epsilon$ |
| (Aspirin) | 200 | $\beta$ | $\beta$ | $\zeta$ |
| | 1000 | $\epsilon$ | $\gamma$ | – |

$(C_{\text{PITC}}, C_{\text{PISC}}, \epsilon) = (0.55, 4.5, 0.01)$.

## D.5. Metric Dependence of PITC

Table A15 provides the result of the metric dependence of PIWSL. For simplicity, we only consider the PITC loss (the coefficient of the PITC and PISC losses are set as 0.1 and 0). For the ReLU metric, we consider

$$L_{\text{ReLU}}(\mathcal{S}; \boldsymbol{\theta}) = \text{ReLU}\left( \left| E(\mathcal{S}; \boldsymbol{\theta}) - \sum_{i=1}^{N_{\text{at}}} \langle \delta \mathbf{r}_i, \mathbf{F}_i(\mathcal{S}; \boldsymbol{\theta}) \rangle - E(\mathcal{S}_{\delta \mathbf{r}}; \boldsymbol{\theta}) \right| - E(\mathcal{S}_{\delta \mathbf{r}}; \boldsymbol{\theta}) \|\delta \mathbf{r}\|^2 \right). \tag{A18}$$

This metric is zero when the difference between the two terms is less than the second-order term in $\delta \mathbf{r}$. The results indicate that taking the second-order term into account does not improve the performance (see PITC MAE Loss and PITC ReLU Loss results), and the MSE loss function shows the best performance. In this study, we used the ANI-1x dataset and the 1000 training samples. These samples differ from the one used to train the model in the main text to avoid overfitting the test dataset.

## D.6. Perturbation Magnitude Dependence

In this section, we provide the result of the perturbation magnitude dependence of PIWSL, i.e., $\|\delta \mathbf{r}\| = \epsilon$. First, we consider only the 1st-order PITC loss (the coefficient of the PITC and PISC losses are set as 0.1 and 0.0). The results are provided in Table A16 and demonstrate that the longer perturbation vector length is fruitful for force predictions. However, values that are too large are harmful to predicting potential energy. In this study, we used the ANI-1x dataset and the 1000 training samples. These samples differ from the one used to train the model in the main text to avoid overfitting the test dataset.

Second, we examine the sensitivity of the second-order PIWSL in the finetuning of MACE-OFF on the MD17 (CCSD) dataset, as discussed in section 5.4. The results, presented in Table A17, indicate that the model's performance is not strongly affected by the choice of $\epsilon_{\text{max}}$, even in the second-order setting.

Table A14: **Energy and force errors for the models employed in the MD analysis.** The presented numerical values are the root-mean-square errors (RMSEs) of energy (E) and force (F). Energy RMSE is given in kcal/mol, while force RMSE is in kcal/mol/Å. "GF" denotes the gradient-based force prediction.

| | | $N_{\text{train}} = 100$ | | $N_{\text{train}} = 200$ | | $N_{\text{train}} = 1000$ | |
| | | Baseline | PIWSL | Baseline | PIWSL | Baseline | PIWSL |
|---|---|---|---|---|---|---|---|
| PaiNN-DF | E | 6.55 | 5.64 | 5.11 | 4.48 | 2.30 | 0.99 |
| | F | 7.38 | 7.36 | 3.95 | 3.97 | 1.63 | 1.61 |
| Equiformer v2 | E | 4.79 | 4.64 | 4.92 | 4.82 | 1.39 | 1.19 |
| | F | 4.86 | 4.90 | 2.50 | 2.42 | 0.76 | 0.75 |
| PaiNN-GF | E | 6.05 | 6.03 | 6.01 | 6.02 | – | – |
| | F | 6.41 | 6.33 | 3.50 | 3.53 | – | – |

Table A15: **Metric dependence of PITC.** The presented numerical values are the root mean square errors (RMSEs) for the ANI-1x dataset (Smith et al., 2020). Energy (in kcal/mol) and force (in kcal/mol/Å) errors are obtained by averaging over three independent runs. All models are trained using 1000 configurations. MAE refers to the mean absolute error, and MSE denotes the mean square error.

| Model | | Baseline | PITC MAE Loss | PITC MSE Loss | PITC ReLU Loss |
|---|---|---|---|---|---|
| PaiNN | E | 60.11 | 58.84 | **47.09** | 60.47 |
| | F | 13.10 | 13.18 | **12.19** | 13.06 |

## D.7. Dependence of PITC on the Number of Perturbed Atoms

This section provides the result of the perturbed atom number dependence of PIWSL. For simplicity, we only consider the PITC loss (the coefficient of the PIRC and PISC losses are set as $0.1$ and $0$). In this study, we randomly selected atoms in a training sample following the ratio of $0\%, 10\%, 20\%, 50\%, 75\%, 90\%, 100\%$. The results are provided in Table A18, which indicates that around 75% to 100% ratio cases result in the best performance for the force and the potential energy prediction. However, the number dependence is rather complicated. Therefore, in the main text, we perturbed all the atoms ($100\%$) as a conservative choice. In this study, we used the ANI-1x dataset and the 1000 training samples. These samples differ from the one used to train the model in the main text to avoid overfitting the test dataset.

## D.8. Dependence on the Number of Training Iterations

To show the effectiveness of our approach even in the case of longer training, we provide the result of the dependence of PIWSL on the number of training iterations. In this study, we performed training twice as long as in the main text, that is, 80,000 iterations for ANI-1x with 1000 training samples. The results are provided in Table A19 and indicate that our approach performs better in the longer training case. On the other hand, the training without PIWSL shows an overfitting to the validation dataset, reducing its performance compared to the shorter training case. In this study, we used the ANI-1x dataset and the 1000 training samples. These samples differ from the one used to train the model in the main text to avoid overfitting the test dataset. The coefficients of the PITC and PISC losses are $1.2$ and $0.01$, respectively.

In Table A20, we also present the performance dependence on training length (measured in epochs) of MACE-OFF model finetuned on MD17 aspirin data with CCSD label. The results demonstrate that PIWSL achieves substantially better performance than the baseline, even when the latter is trained for ten times as many epochs.

## D.9. Additional Experiments with Gradient-Based Forces

In this section, we provide the result of the training with the gradient-based force predictions. The results are provided in Table A21 and demonstrate that our PIWSL loss enables a better force prediction, even in the case of gradient-based force predictions. These results also indicate that our PIWSL method can improve the ML model performance in the case of MLIPs commonly applied in computational chemistry and materials science. We consider that this is partly due to the

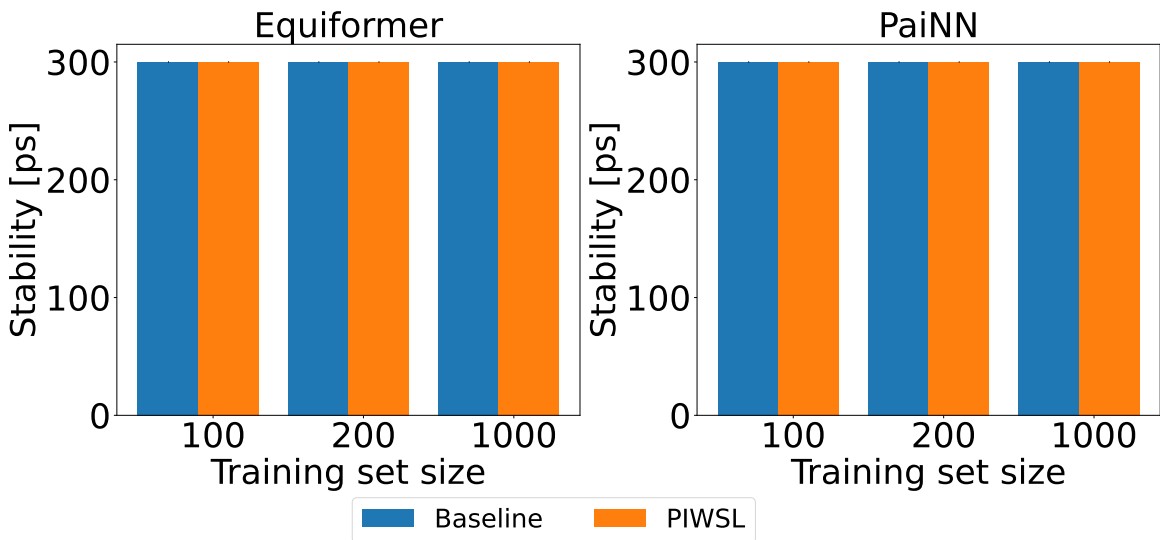

Figure A1: **Stability analysis of the MLIP models during MD simulations.** Stability during MD simulations is assessed for the baseline MLIP models and those trained with PIWSL. All results are obtained for the aspirin molecule and MD simulations in the canonical (N V T) statistical ensemble. We measure stability during MD simulations according to (Fu et al., 2023).

Table A16: **Perturbation magnitude dependence of PITC.** The presented numerical values are the root mean square errors (RMSEs) for the ANI-1x dataset (Smith et al., 2020). Energy (in kcal/mol) and force (in kcal/mol/Å) errors are obtained by averaging over three independent runs. All models are trained using 1000 configurations.

| Model | | Baseline | $\epsilon_{\max} = 0.0005$ | $\epsilon_{\max} = 0.005$ | $\epsilon_{\max} = 0.05$ |
|---|---|---|---|---|---|
| PaiNN | E | 60.11 | 60.43 | **47.09** | 109.17 |
| | F | 13.10 | 12.75 | 12.19 | **11.70** |

effectiveness of the weak label at $\mathbf{r} + \delta\mathbf{r}$ as indicated by the WL results, which show an improvement of the performance different from the case with the direct force branch (see also section E). We hypothesize that the further improvement results from the additional gradient calculation as indicated in Eq. (A5) and Eq. (A6). This observation also indicates that our PIWSL method can potentially improve other generic property prediction tasks by calculating their first derivatives in terms of the atomic coordinate and utilizing the proposed loss functions. In this study, the coefficient of the PITC and PISC losses are set as 0.1 and 0.01 with $\epsilon_{\max} = 0.01$. The weak label loss coefficient is set as 0.1.

### D.10. Reducing Curl of Forces for Models with the Force Branch

In this section, we study the effect of our loss functions on the curl of forces in the case of the model with the force branch. The results are provided in Table A22, which shows that our PITC loss reduces the curl of the predicted forces, allowing potentially better energy conservation during MD simulations. In this study, we used the ANI-1x dataset and the 1000 training samples. These samples differ from the one used to train the model in the main text to avoid overfitting the test dataset. The hyper-parameters of the PITC and PISC losses are $(C_{\mathrm{PITC}}, C_{\mathrm{PISC}}, \epsilon_{\max}) = (1.2, 0.01, 0.025)$. It is theoretically possible to define a loss function aimed at directly minimizing the absolute value of the curl of forces. However, this approach necessitates calculating the Hessian matrix, which requires a substantial memory cost given the limitations of current computational resources. Developing a method to train with such a loss function while mitigating memory requirements is a promising direction for future research.

Table A17: **Performance dependence of the second order PIWSL on** $\epsilon_{\max}$**.** The numerical values are MAEs for the energy in kcal/mol and force in kcal/mol/Å of MACE-OFF model finetuned on MD17-CC dataset.

| Model | | Baseline | $\epsilon_{\max} = 0.005$ | $\epsilon_{\max} = 0.01$ | $\epsilon_{\max} = 0.02$ |
|---|---|---|---|---|---|
| MACE-OFF | E | 1.21 | 0.73 | **0.72** | 0.75 |
| | F | 6.90 | 4.20 | **3.77** | 4.04 |

Table A18: **Dependence of PITC on the number of perturbed atoms.** The presented numerical values are the root mean square errors (RMSEs) for the ANI-1x dataset (Smith et al., 2020). Energy (in kcal/mol) and force (in kcal/mol/Å) errors are obtained by averaging over three independent runs. All models are trained using 1000 configurations.

| Model | | Baseline | 10% | 20% | 50% | 75% | 90% | 100 % |
|---|---|---|---|---|---|---|---|---|
| PaiNN | E | 60.11 | 46.68 | 52.37 | 54.51 | 46.94 | **45.92** | 46.32 |
| | F | 13.10 | 13.03 | 12.62 | 12.16 | **12.14** | 12.24 | 12.42 |

## E. Further Analyses of PIWSL

The following provides further analyses of our approach. We provide the results for Equiformer v2 and PaiNN since these models employ equivariant features and demonstrate a high accuracy on the ANI-1x dataset when trained using 1000 configurations.

### E.1. Training MLIPs without Reference Forces

In the following, we explore scenarios where only potential energy labels are available. This situation commonly arises when calculating energy labels with chemically accurate approaches, such as CCSD(T)/CBS (Hobza & Šponer, 2002; Feller et al., 2006), for which force calculation is infeasible. To consider practical applications, we examine two cases: (1) predicting force by a force branch (FB) and (2) predicting force as a gradient of the potential energy (GF). The former enables fast force prediction and is popular in the machine learning community, while the latter requires additional gradient calculation but yields curl-free force predictions. It is popular in computational chemistry as it ensures the conservation of the total energy during MD simulations. The results are provided in Table A23; training without reference forces is achieved by setting the relative force contribution to zero in Eq. (1). The PIWSL method consistently performs better than the baseline for the FB and GF cases. However, a more significant improvement in the force prediction performance is observed in the GF case. We attribute this phenomenon to the inherent nature of PIWSL, which requires consistency between the potential energy and atomic forces, as discussed in section 5.3. This result aligns with our expectations, confirming the capability of our PIWSL method to enable ML models to reduce the error in the predicted forces. Overall, PIWSL opens a new possibility for training MLIP models using highly accurate reference methods, such as CCSD(T)/CBS.

### E.2. Comparing PITC with the Taylor-Expansion-Based Weak Label Loss.

We compare the PIWSL method with the Taylor-expansion-based weak label (WL) approach (Cooper et al., 2020), whose loss function is presented in Eq. (A3). For simplicity, we only consider the PITC loss in Eq. (3). For a fair comparison, we consider the following two cases. First, we train with reference forces and energies (w. RF). Second, we train the methods without reference forces and use only the reference energies. For the training with reference forces, we set the numeric coefficient of the PITC loss to 1.0; for the training without reference forces, the coefficient is set to 0.1. Note that the WL loss without the reference force is calculated using the predicted force labels. The results are provided in Table A24.

Our PITC loss demonstrates the best accuracy in all cases, with and without the reference forces. Interestingly, PaiNN failed to learn the potential energy with the WL loss and reference forces. We hypothesize this to be due to the imbalance of the training between the energies and forces. Specifically, the WL loss trains only the potential energy, resulting in an inconsistency between the energy and force branches, which share the same readout layer that experiences more frequent updates using the potential energy. This hypothesis is supported by the results for the training without reference forces, where the error in energy is reduced compared to the baseline. A further validation in a similar experiment in the case of GF

Table A19: **Dependence on the number of training iterations.** The presented numerical values are the root mean square errors (RMSEs) for the ANI-1x dataset (Smith et al., 2020). Energy (in kcal/mol) and force (in kcal/mol/Å) errors are obtained by averaging over three independent runs. All models are trained using 1000 configurations.

| Model | Iteration Number | | Baseline | PIWSL |
|---|---|---|---|---|
| PaiNN | 40,000 | E | $56.62 \pm 2.80$ | $\mathbf{24.53 \pm 0.16}$ |
| | | F | $12.96 \pm 0.18$ | $\mathbf{11.43 \pm 0.05}$ |
| | 80,000 | E | $59.92 \pm 1.47$ | $\mathbf{23.78 \pm 0.16}$ |
| | | F | $13.10 \pm 0.19$ | $\mathbf{11.50 \pm 0.04}$ |

Table A20: **Dependence on the number of training iterations of MACE-OFF finetuning.** The presented numerical values are the mean absolute errors (MAEs) for the MD17-CC, in terms of energy (in kcal/mol) and force (in kcal/mol/Å).

| Model | Epoch Number | | Baseline | PIWSL |
|---|---|---|---|---|
| MACE-OFF | 100 | E | 1.21 | **0.72** |
| | | F | 6.90 | **3.77** |
| | 1,000 | E | 1.19 | – |
| | | F | 6.24 | – |

is provided in section D.9. However, the proposed PITC loss still performs better here. In summary, the PITC loss enables MLIPs to learn energies and forces consistent with each other and does it better than the previously proposed WL method.

### E.3. Ablating the Impact of PITC and PISC Losses.

We conduct an ablation experiment to analyze the impact of PITC and PISC losses. Results in Table A25 indicate that the PITC loss predominantly improves the accuracy of resulting models, especially for PaiNN. Using just the PISC loss does not consistently improve accuracy but stabilizes training when combined with PITC. This combined approach notably benefits Equiformer v2. For Equiformer v2, we repeated the experiment five times to reduce the effect from an outlier on the PITC loss.

### E.4. Adversarial Directions for Perturbing Atomic Positions.

The following discusses the dependence of the PIWSL's performance on selecting the vector $\delta \mathbf{r}$ in Eq. (5) employed to perturb atomic positions. The detailed implementation and setups are provided in section B.1. Table A26 compares the results obtained for a randomly-sampled vector $\delta \mathbf{r}$ and for the one determined adversarially. The results demonstrate that both approaches improve the performance compared to the baseline without weak supervision, though the results might depend on the employed model.

### E.5. Performance Difference Between First and Second order PIWSL

In this section, we present a comparison between first- and second-order PIWSL methods. We evaluate a foundation model, MACE-OFF, fine-tuned on the MD17 Aspirin dataset with CCSC(T) labels and the MD22 buckyball catcher dataset, as well as PaiNN-GF, a gradient-based force model trained on 1,000 samples from ANI-1x. The results are provided in Table A27.

Our findings indicate that the second-order term becomes more significant as model performance improves, particularly when force prediction error becomes small. This suggests that the first-order term, which encourages the model to predict the potential energy of neighboring conformations based on a first-order Taylor expansion, plays a crucial role in promoting a smoother potential energy surface. This effect arises because, in MACE and PaiNN-GF models, force predictions are derived as the gradient of the potential energy surface.

Table A21: **Results of PIWSL with gradient-based force predictions.** The presented numerical values are the root mean square errors (RMSEs) for the ANI-1x dataset (Smith et al., 2020). Energy (in kcal/mol) and force (in kcal/mol/Å) errors are obtained by averaging over three independent runs. All models are trained using 1000 configurations.

| Model | | Baseline (GF) | PIWSL (GF) | WL (GF) |
|---|---|---|---|---|
| PaiNN | E | $23.57 \pm 0.62$ | $\mathbf{18.62 \pm 0.09}$ | $22.61 \pm 0.50$ |
| | F | $11.32 \pm 0.08$ | $\mathbf{10.94 \pm 0.01}$ | $11.72 \pm 0.06$ |
| Equiformer | E | $29.07 \pm 2.32$ | $\mathbf{19.53 \pm 0.32}$ | $21.07 \pm 0.86$ |
| | F | $\mathbf{11.90 \pm 0.13}$ | $11.99 \pm 0.03$ | $11.90 \pm 0.20$ |

Table A22: **Curl of forces for models with the force branch.** The presented numerical values are the absolute values of the total curl of the force evaluated for the ANI-1x dataset (Smith et al., 2020). Energy (in kcal/mol) and force (in kcal/mol/Å) errors are obtained by averaging over three independent runs. All models are trained using 1000 configurations.

| Model | Baseline | PITC |
|---|---|---|
| PaiNN | $45.18 \pm 4.07$ | $\mathbf{39.06 \pm 0.58}$ |
| Equiformer | $29.62 \pm 0.28$ | $\mathbf{23.42 \pm 0.09}$ |

Table A23: **Results for models trained on the ANI-1x dataset without reference forces.** All models are trained using 1000 training samples. FB refers to the setting where the force branch estimates the force, and GF denotes the setting where the force is estimated by the gradient of the potential energy with respect to the atomic coordinates.

| Model | Case | | Baseline | PIWSL |
|---|---|---|---|---|
| PaiNN | FB | E | $42.36 \pm 0.30$ | $\mathbf{25.42 \pm 0.72}$ |
| | | F | $24.25 \pm 0.00$ | $\mathbf{20.54 \pm 0.08}$ |
| | GF | E | $41.83 \pm 1.81$ | $\mathbf{29.71 \pm 0.55}$ |
| | | F | $83.36 \pm 2.85$ | $\mathbf{24.02 \pm 0.95}$ |
| Equiformer | FB | E | $43.14 \pm 0.86$ | $\mathbf{29.48 \pm 0.51}$ |
| | | F | $24.25 \pm 0.00$ | $\mathbf{21.99 \pm 0.49}$ |
| | GF | E | $42.55 \pm 0.99$ | $\mathbf{32.66 \pm 1.11}$ |
| | | F | $35.70 \pm 0.78$ | $\mathbf{21.83 \pm 0.27}$ |

Table A24: **Comparison of PITC and the Taylor-expansion-based weak label loss.** WL (+FP) denotes the Taylor-expansion-based method using reference energies and either reference (w. RF) or predicted (w/o. RF) forces; see Eq. (A3). The listed values are the RMSE values for energies in kcal/mol and atomic forces in kcal/mol/Å. All models are trained on the ANI-1x dataset using 1000 configurations, with (w.) and without (w/o.) reference atomic forces (RF).

| Model | Case | | Baseline | PITC | WL (+FP) |
|---|---|---|---|---|---|
| PaiNN | (w. RF) | E | $56.62 \pm 2.80$ | $\mathbf{30.94 \pm 0.56}$ | $81.86 \pm 9.39$ |
| | | F | $12.96 \pm 0.06$ | $\mathbf{12.04 \pm 0.04}$ | $14.54 \pm 0.12$ |
| | (w/o. RF) | E | $42.36 \pm 0.30$ | $\mathbf{25.42 \pm 0.72}$ | $41.77 \pm 4.82$ |
| | | F | $24.25 \pm 0.00$ | $\mathbf{20.54 \pm 0.08}$ | $24.68 \pm 0.54$ |
| Equiformer | (w. RF) | E | $54.52 \pm 4.52$ | $\mathbf{23.16 \pm 0.19}$ | $31.02 \pm 3.99$ |
| | | F | $10.10 \pm 0.00$ | $\mathbf{10.03 \pm 0.05}$ | $13.43 \pm 0.92$ |
| | (w/o. RF) | E | $43.14 \pm 0.86$ | $\mathbf{29.48 \pm 0.51}$ | $88.59 \pm 11.36$ |
| | | F | $24.25 \pm 0.00$ | $\mathbf{21.99 \pm 0.49}$ | $293.41 \pm 26.96$ |

Table A25: **Results for models with direct-force trained on the ANI-1x dataset with ablated weakly supervised losses.** All models are trained using 1000 training samples. All results are obtained by averaging over three independent runs. Energy RMSE is given in kcal/mol, while force RMSE is given in kcal/mol/Å.

| Model | PITC | PISC | E | F |
|---|---|---|---|---|
| PaiNN | ✗ | ✗ | $56.62 \pm 2.80$ | $12.96 \pm 0.06$ |
| | ✓ | ✗ | $\mathbf{24.60 \pm 0.18}$ | $\mathbf{11.51 \pm 0.03}$ |
| | ✗ | ✓ | $58.30 \pm 2.10$ | $13.18 \pm 0.29$ |
| | ✓ | ✓ | $\mathbf{24.53 \pm 0.48}$ | $\mathbf{11.43 \pm 0.05}$ |
| Equiformer | ✗ | ✗ | $54.52 \pm 4.52$ | $10.10 \pm 0.00$ |
| | ✓ | ✗ | $32.64 \pm 26.48$ | $\mathbf{9.64 \pm 0.03}$ |
| | ✗ | ✓ | $48.96 \pm 4.96$ | $10.30 \pm 0.06$ |
| | ✓ | ✓ | $\mathbf{20.89 \pm 0.50}$ | $\mathbf{9.68 \pm 0.03}$ |

Table A26: **PIWSL's performance dependence on the atomic position perturbation vector.** The numerical values are RMSEs for the energy in kcal/mol and force in kcal/mol/Å. All results are provided for the ANI-1x dataset and models trained using 1000 configurations.

| | | Baseline | Random (Eq. (6)) | Adversarial (Eq. (7)) |
|---|---|---|---|---|
| PaiNN | E | $56.62 \pm 2.80$ | $\mathbf{24.53 \pm 0.48}$ | $33.67 \pm 1.12$ |
| | F | $12.96 \pm 0.18$ | $\mathbf{11.43 \pm 0.05}$ | $12.74 \pm 0.14$ |
| Equiformer | E | $54.52 \pm 4.52$ | $23.16 \pm 0.50$ | $\mathbf{20.54 \pm 0.21}$ |
| | F | $10.10 \pm 0.00$ | $10.03 \pm 0.03$ | $\mathbf{9.93 \pm 0.04}$ |

Table A27: **Performance difference between first and second order PIWSL.** The numerical values are RMSEs (ANI-1x) and MAE (MD17-CC/MD22-BB) for the energy in kcal/mol and force in kcal/mol/Å. MD17-CC denotes MD17 aspirin data with CCSD label, MD22-BB denotes MD22 buckyball catcher data, and ANI-1x (1K) denotes 1000 samples from ANI-1x.

| Dataset | Model | | Baseline | 1st-order PIWSL | 2nd-order PIWSL |
|---|---|---|---|---|---|
| MD17-CC | MACE-OFF | E | $1.21 \pm 0.00$ | $0.90 \pm 0.03$ | $\mathbf{0.72 \pm 0.01}$ |
| | | F | $6.90 \pm 0.01$ | $4.59 \pm 0.16$ | $\mathbf{3.77 \pm 0.13}$ |
| MD22-BB | MACE-OFF | E | $1.16 \pm 0.15$ | $1.16 \pm 0.28$ | $\mathbf{0.99 \pm 0.05}$ |
| | | F | $0.35 \pm 0.00$ | $0.35 \pm 0.00$ | $\mathbf{0.34 \pm 0.00}$ |
| ANI-1x (1K) | PaiNN-GF | E | $23.57 \pm 0.62$ | $\mathbf{18.62 \pm 0.09}$ | $18.88 \pm 0.23$ |
| | | F | $11.32 \pm 0.08$ | $\mathbf{10.94 \pm 0.01}$ | $11.01 \pm 0.00$ |

