# OpenReview forum: "Physics-Informed Weakly Supervised Learning For Interatomic Potentials"
_ICML.cc/2025/Conference — ICML 2025 poster_

### Official Review · Reviewer_gwDq · 2025-03-07

**Overall Recommendation:** 4

**Summary:**

The paper proposes a new method for training machine learning interatomic potentials (MLIPs) using loss functions based on the Taylor expansion of the potential energy and the notion of conservative forces. The papers starts by describing ab-initio computational chemistry simulation methods and motivates the need for MLIPs as tool to bridge ab-initio methods and classical force fields. Next, the paper describes data generation as one of the major challenges for MLIP training given the high computational cost of computational chemistry methods that serve as the primary source of MLIP data. Furthermore, the authors claim that MLIPs have trouble generalizing and claim that their methods helps mitigate such challenges. The proposed method, physics-informed weakly supervised learning (PIWSL), is based on two physics-informed losses: physics-informed Taylor-expansion-based consistency (PITC) and physics-informed spatial consistency (PISC). The authors claim that MLIPs trained with PIWSL require less training data, produce more accurate energy prediction and increases robustness of MLIPs.

Section 2 covers related work on MLIPs, focusing on architectures and methods, and physics-informed machine learning covering various approaches to infuse physics principles into machine learning methods. Section 3 outlines relevant background on training MLIPs, including the common training loss definition centered on energy and force labels. Section 3 also introduces weakly supervised learning for MLIPs, which relies on training on structures with small perturbations. Section 4 describes the details of weakly supervised learning focusing on the two PIWSL losses. The PITC takes the Taylor expansion of the energy a perturbed structure and re-writes it as the inner product of forces and positions. PITC also introduces a parameter that modulates the contribution of the second order, thereby enabling training with different contributions of the force field terms. PISC, the second loss, leverages the conservative nature of forces to create weak supervision. The method leverages perturbations by different paths to the same perturbed state to enforce a consistency loss. The paper then combines the two losses into a joint loss and discusses considerations for perturbation directions and magnitudes.

Section 5 outlines the experiments starting with energy and force error experiments on ANI-1x and TiO2 while concurrently ablating dataset size. The experiments ablate five different models and compares results to noisy nodes pre-training for different dataset sizes. The results in Table 1 and Table 2 generally show that PIWSL outperformance baseline training and noisy nodes for most of the cases studied. Section 5.3 details a qualitative assessment of the effects of PIWSL for the aspirin molecule which claim greater prediction improvement and stability in MD simulations. Section 5.4 outlines experiments related to fine-training pretrained MLIPs, such as MACE, with a regular training loss compared to PIWSL. The results generally show that MACE trained with PIWSL performs better than baseline losses both for pretrained and randomly initialized MACE models. Section 6 provides a conclusion and brief discussion of limitations.

## update after rebuttal

The authors answered my questions to my satisfaction and agreed to include relevant details that will make the paper stronger. These include details on limitations, computational cost trade-offs and additional details on comparing with DeNS. As such, I maintain my support of the paper.

**Claims And Evidence:**

The claims made in the paper, that PIWSL based training requires less data and improves MLIP performance, are generally well supported with the experiments shown in Section 5. The claim related to increased robustness in MD simulation has some supporting evidence, but is mainly on a single case study, and could be further strengthened with additional evidence.

Furthermore, the paper mainly shows experiments on molecular structures (outside of the TiO2 dataset). The claims and evidence could be further strengthened by providing studies of solid-state structures (examples datasets can be found here [1] [2]). At the minimum, the limitations should mention that future work is needed to assess PIWSL for a broader set of materials.

[1] Bihani, V., Mannan, S., Pratiush, U., Du, T., Chen, Z., Miret, S., Micoulaut, M., Smedskjaer, M.M., Ranu, S. and Krishnan, N.A., 2024. EGraFFBench: evaluation of equivariant graph neural network force fields for atomistic simulations. Digital Discovery, 3(4), pp.759-768.

[2] Lee, K.L.K., Gonzales, C., Nassar, M., Spellings, M., Galkin, M. and Miret, S., MatSciML: A Broad, Multi-Task Benchmark for Solid-State Materials Modeling. In AI for Accelerated Materials Design-NeurIPS 2023 Workshop.

**Essential References Not Discussed:**

I would recommend the paper discuss the papers mentioned in the previous boxes. [1] is especially important for the MD analysis section as it provides relevant metrics and analysis techniques for that case. [2] discusses datasets relevant for MLIPs for solid-state materials and [3] outlines a different pre-training method.

On top of that, [4] is a useful review of model architectures and methods related to atomistic modeling that would be useful to provide a broader perspective on MLIPs and geometric deep learning models used for property prediction. A recent paper [5] might also be worth mentioned as it reinforces many of the arguments made in the paper related to costly training data and scaling MLIPs to more difficult challenges.

[1] Bihani, V., Mannan, S., Pratiush, U., Du, T., Chen, Z., Miret, S., Micoulaut, M., Smedskjaer, M.M., Ranu, S. and Krishnan, N.A., 2024. EGraFFBench: evaluation of equivariant graph neural network force fields for atomistic simulations. Digital Discovery, 3(4), pp.759-768.

[2] Lee, K.L.K., Gonzales, C., Nassar, M., Spellings, M., Galkin, M. and Miret, S., MatSciML: A Broad, Multi-Task Benchmark for Solid-State Materials Modeling. In AI for Accelerated Materials Design-NeurIPS 2023 Workshop.

[3] Liao, Y.L., Smidt, T., Shuaibi, M. and Das, A., Generalizing Denoising to Non-Equilibrium Structures Improves Equivariant Force Fields. Transactions on Machine Learning Research.

[4] Duval, A., Mathis, S.V., Joshi, C.K., Schmidt, V., Miret, S., Malliaros, F.D., Cohen, T., Liò, P., Bengio, Y. and Bronstein, M., 2023. A hitchhiker's guide to geometric gnns for 3d atomic systems. arXiv preprint arXiv:2312.07511.

[5] Miret, S., Lee, K.L.K., Gonzales, C., Mannan, S. and Krishnan, N.M., 2025. Energy & Force Regression on DFT Trajectories is Not Enough for Universal Machine Learning Interatomic Potentials. arXiv preprint arXiv:2502.03660.

**Experimental Designs Or Analyses:**

I checked the experiments and analyses presented in the main paper and some of the additional details in the supplementary material.

**Methods And Evaluation Criteria:**

The methods and evaluation criteria generally make sense for the claims presented. The varying of dataset size in Section 5.2 is particularly useful for supporting the data efficiency claims and providing ideas for future research. As mentioned in the previous in the previous box, the paper could be improved by discussing the limitations of the benchmark used and how it relates to the broad set of materials and molecules that MLIPs aim to cover.

One additional baseline that would be useful to include, or discuss at the minimum, is DeNS [1], which also outlines a pre-training method based on perturbed non-equilibrium structures.

[1] Liao, Y.L., Smidt, T., Shuaibi, M. and Das, A., Generalizing Denoising to Non-Equilibrium Structures Improves Equivariant Force Fields. Transactions on Machine Learning Research.

**Other Comments Or Suggestions:**

N/A

**Other Strengths And Weaknesses:**

Overall the paper provides an interesting methodological innovation that could be broadly applicable to MLIP training and provides solid evidence for its claims.

**Strengths:**
* Interesting methodological innovation based on physical principles that can help alleviate data sparsity for MLIP training.
* The proposed method is model agnostic and hence potentially broadly applicable.
* The experiments provide support for the claims around data efficiency.

**Weaknesses:**
* The paper could be strengthened by discussing broader applications of MLIPs and ideally conducting more experiments on different systems to show the applicability of PIWSL. At a minimum, limitations should more clearly outline what the experiments show and what is future work.
* The mentions a training time analysis in Appendix D that states that PIWSL leads to increased training time. It would be worth mentioning this in the main paper as a trade-off.

**Questions For Authors:**

1. What is the reason you focused most of your experiments and analysis on molecular systems and single molecules? Do you believe this provides enough evidence for the benefits of PIWSL? Why or why not?
2. What extension of PIWSL do you think could strengthen MLIP training (e.g., third order Taylor expansion or other weak supervision)? How does that trade off with additional computational cost?
3. Do you think PIWSL representations could be useful for property prediction tasks?

**Relation To Broader Scientific Literature:**

The contributions and findings provide a new useful formulation and insights for the training of MLIPs. As mentioned in prior boxes, the paper could be strengthened by discussing MLIP application beyond molecular structures.

**Theoretical Claims:**

The theoretical claims in the paper are generally well supported. I would recommend re-writing Equation 2 to include relevant gradient/derivatives that relate energy and force to make it easier for the reader to follow.

---

> ### Author Rebuttal · Authors · 2025-03-31
>
> Dear Reviewer,
>
> Thank you for your constructive and encouraging feedback. We have carefully considered your comments and will incorporate the necessary revisions in the camera-ready version. Below, we provide detailed responses to each of your encouraging proposals and questions. Please let us know if any of your concerns remain unaddressed.
>
> $\\textbf{Q1}$. We agree that evaluating PIWSL on additional datasets would be beneficial. However, we have evaluated PIWSL across various MLIP models and datasets, including heterogeneous datasets such as ANI-1x. As recommended, we will discuss the limitations of the benchmark dataset in the "Limitations" section and plan to explore a broader range of datasets in future work.
>
> $\\textbf{W2/Q2}$. While a third-order extension may improve accuracy, the computational cost of calculating second and third-order derivatives of the potential may outweigh the benefits. An alternative approach would be to explore new PISC spatial configurations, which offer greater flexibility than PITC, as discussed in Appendix E.2.
> We will mention the computational cost trade-off mentioned in Weakness 2 in the main-body in the camera-ready version. The performance improvements achieved by PIWSL justify the additional computational cost. For example, Table A20 compares training with extended iterations, showing that while the baseline method overfits the training data, PIWSL continues to improve. This is particularly important for the sparse data regime, which is our primary focus. Moreover, the increase in training time is negligible compared to the substantial computational cost and time required for data generation, particularly when using DFT or coupled-cluster methods. This becomes even more significant as molecular size increases.
>
> $\textbf{Q3}$. Thank you for your insightful comment! PITC is applicable whenever the target property depends on atomic coordinates. For instance, DeNS, the recommended work by the reviewer, applied their method to relaxation energy estimation in OC20/22, and we expect that PIWSL could also be effectively applied to this work. We are excited about the potential to explore new application domains in future research.
>
> $\textbf{Comparison with DeNS (Liao+2024)}$
>
> We appreciate the recommendation of DeNS. We acknowledge that DeNS was recently accepted on December 27th. Although we have already cited it in Appendix C.2 as a related work, we will provide a more detailed discussion of these differences in the “Related Work” section of the main body in the camera-ready version.
>
> The key differences between PIWSL and DeNS are as follows:
>
> 1) DeNS requires force labels, whereas PIWSL does not. This allows PIWSL to fine-tune MLIP models using datasets containing only potential energies, as demonstrated in Table 3. This scenario is particularly interesting for methods like CC/CBS.
>
> 2) While DeNS introduces a smaller increase in training time, PIWSL offers greater flexibility by eliminating the need for force supervision, making it more broadly applicable in scenarios where force labels are unavailable.
>
> 3) DeNS incorporates an equivariant force encoding module within the target MLIP model, which requires integration into the training pipeline. In contrast, PIWSL directly refines energy-based models without such modifications.
>
> 4) DeNS primarily discusses energy and force errors but does not explicitly discuss robustness in molecular dynamics (MD) simulations, which is one of the key aspects in our work.
>
> Additionally, an interesting open question is whether techniques from DeNS could complement PIWSL to further enhance MLIP performance. While we have not explored this combination in this work, investigating such synergies could be valuable for future research.

---

> > ### Comment · Reviewer_gwDq · 2025-04-03
> >
> > Thank you for the additional details. I am comfortable maintaining my score and support of the paper. I think including the additional cost-performance trade-off for PIWSL described in W2/Q2 is particularly helpful.

---

> > > ### Author Response · Authors · 2025-04-03
> > >
> > > Thank you for your supportive feedback. We truly appreciate your insights and will ensure that the cost-performance trade-off discussion is included in the camera-ready version if our paper is accepted.
> > >
> > > Authors

---

### Official Review · Reviewer_Q1cJ · 2025-03-14

**Overall Recommendation:** 3

**Summary:**

The paper proposes a physics-informed weakly supervised learning (PIWSL) framework to improve the accuracy and robustness of machine-learned interatomic potentials (MLIPs). PIWSL incorporates two new loss functions: Physics-Informed Taylor-Expansion-Based Consistency (PITC) loss and Physics-Informed Spatial Consistency (PISC) loss. These losses help refine energy and force predictions, particularly when training data is sparse. The paper demonstrates significant improvements in predictive accuracy and robustness across various molecular and material datasets. Additionally, PIWSL enhances the fine-tuning of foundation models.

**Claims And Evidence:**

The claims in the paper are generally well-supported with empirical evidence. The authors provide extensive benchmarking on diverse datasets, including ANI-1x, TiO2, and MD17(CCSD), showcasing improvements in accuracy. They also validate robustness through MD simulations. Theoretical justifications, particularly regarding the formulation of loss functions, are sound. However, further analysis of the trade-offs in computational cost associated with PIWSL would strengthen the claims.

**Essential References Not Discussed:**

The paper presents a solid literature review. However, existing works that utilize weak supervision signals for training MLIPs [1] may offer a more comprehensive review.

[1] Shui, Zeren, et al. "Injecting domain knowledge from empirical interatomic potentials to neural networks for predicting material properties." Advances in Neural Information Processing Systems 35 (2022): 14839-14851.

**Experimental Designs Or Analyses:**

The experimental design is appropriate. The authors apply the proposed training method to multiple datasets and baseline models. The authors also conduct experiments to study the impact of PIWSL on training set sizes, and the robustness during MD simulations.

**Methods And Evaluation Criteria:**

The proposed methods are well-aligned with the problem of training MLIPs. The authors employ relevant benchmark datasets, including molecular and material science datasets, and compare their approach with state-of-the-art methods such as SchNet, PaiNN, and Equiformer. Evaluation criteria include RMSE and MAE metrics for energy and force predictions, along with robustness assessments via MD simulations. The experimental setup is comprehensive and appropriately chosen for the task.

**Other Comments Or Suggestions:**

Typo
Line 232, "nergy" -> "energy".

**Other Strengths And Weaknesses:**

Strengths:

1. The proposed method is innovative and effective in improving the accuracy and robustness of MLIPs in the data sparse scenario.

2. The proposed method outperforms existing methods such as NoisyNodes on a wide range of benchmark datasets and base models.

Weaknesses:

1. Limited discussion on computational costs and efficiency.

2. The method’s improvement diminishes as the training data size increases.

**Questions For Authors:**

1. Does the method lead to unstable training in the initial stages when the neural networks are randomly initialized? The predicted energy and forces used to compute the supervision signal may deviate significantly.

2. In Table 2/3, why does NoisyNodes significantly degrades the performance of the base models?

I will adjust my score once these questions and the first weakness are addressed.

**Relation To Broader Scientific Literature:**

The paper aims to improve the generalizability of MLIPs in tasks where a limited amount of training data is available. In the field of atomistic modeling, data scarcity is a common challenge given the computational complexity of first-principle methods (DFT) or the demanding cost of conducting wet-lab experiments.

**Theoretical Claims:**

The paper provides a theoretical derivation of PITC and PISC loss functions. The use of Taylor expansions to approximate perturbed energy values is mathematically sound.

---

> ### Author Rebuttal · Authors · 2025-03-31
>
> Dear Reviewer,
>
> Thank you for your constructive feedback. We have carefully considered your comments and will incorporate the necessary revisions in the camera-ready version. Below, we provide detailed responses to each point.
>
> $\\textbf{W1}$:
> A detailed discussion regarding the computational time required for both the baseline and PIWSL is provided in Tables A10 and A11 in Appendix E2. Additionally, data sample efficiency is partly analyzed in Table 3, demonstrating that PIWSL achieves comparable or better accuracy while requiring only half the training data for finetuning a foundation model.
> Since our primary focus is on the sparse data regime, as explained in the paper, data efficiency is a critical factor for us. This makes PIWSL particularly valuable in our target setting. We will reference these results more prominently in the main paper.
>
> $\\textbf{W2}$:
> We acknowledge the reviewer’s concern that PIWSL’s relative improvement decreases as dataset size increases. This trend is expected because, in data-rich scenarios, supervised learning already provides strong generalization, reducing the additional benefit of weak labels. However, PIWSL remains valuable, particularly in materials discovery, where data is scarce and MLIP models must be trained from scratch or fine-tuned from foundation models.
> Even in data-rich settings, PIWSL still provides meaningful improvements. For instance, with the ANI-1x 5M dataset, PIWSL achieves over a 10% error reduction, demonstrating that weak labels enhance learning even when abundant supervised data is available. While the marginal gains decrease, the improvements remain significant given the high precision requirements in MLIP applications.
>
> $\textbf{Q1}$: Although training stability depends on the coefficients of PITC and PISC, we do not observe instability in the initial stages of training with PIWSL. This is because, at the beginning of training, the energy and force regression losses are significantly larger than the PIWSL losses. While increasing the PIWSL loss coefficients could lead to instability in the initial phase, it would also degrade overall performance. If instability were to arise, a potential mitigation strategy would be gradually increasing the PIWSL loss weight during training (similar to curriculum learning).
>
> $\textbf{Q2}$: As stated in the final sentence of Section 5.2 ("Heterogeneous Molecular Dataset"), we attribute this issue to NoisyNode's inability to properly capture the response of energy and atomic forces to perturbations in atomic positions.

---

### Official Review · Reviewer_T8Zw · 2025-03-20

**Overall Recommendation:** 3

**Summary:**

In this paper, the authors propose two auxiliary loss functions to improve the generalization of machine learning interatomic potentials (MLIP). Using molecular and crystal datasets, they demonstrate that the proposed method enhances the accuracy of energy and force predictions across multiple MLIPs.

## update after rebuttal
The author's responses addressed my questions. In particular, my major concerns about the convergence of the training of the baseline methods were resolved, so I raised my evaluation. However, I'm not convinced that simply using Taylor expansion is called 'physics informed'. More explanation is needed.

I would like those responses to be reflected in the main text.

**Claims And Evidence:**

The authors claim that the proposed "Physics-Informed" auxiliary loss functions improve the accuracy of MLIPs (Machine Learning Interatomic Potentials). They experimentally demonstrate improvements in the accuracy of various MLIPs on molecular and crystal datasets.

The proposed auxiliary loss functions consist of PITC (Physics-Informed Taylor-Expansion-Based Consistency Loss) and PISC (Physics-Informed Spatial-Consistency Loss).
PITC uses a second-order Taylor approximation to create pseudo-labels for energy, but it does not explain the physical justification. Hence, it does not fully support the "Physics-Informed" claim. PITC might be an analogy with the harmonic oscillator approximation that uses a second-order approximation around stable equilibrium points.
On the other hand, PISC is based on the physical principle that the energy difference is path-independent, and it successfully supports the claim of being "Physics-Informed."

**Essential References Not Discussed:**

I think many readers want to see the comparison with (Liao et al., 2024) since they also improved EquiformerV2 and eSCN on both crystal and molecule datasets.

**Experimental Designs Or Analyses:**

The experimental setup follows existing research. However, there are a few points that are slightly concerning due to a lack of clear explanation:

1. The proposed method takes 2-3 times longer to train compared to the baselines, so I'm concerned whether the advantage over the baseline methods wouldn't be slight if the number of training iterations was increased. There would be no issue if it could be confirmed that the baseline methods have sufficiently converged with the current number of training iterations.
2. Since the proposed method is expected to be sensitive to the value of $\epsilon$, I would like to ensure that the value of epsilon was not cherry-picked.

**Methods And Evaluation Criteria:**

Since generating training data for MLIP is time-consuming, there is a need to train or fine-tune with a small amount of training data, and the proposed method addresses this issue. Moreover, evaluating the accuracy of energy and forces on molecular and crystal datasets is a fundamental and appropriate method of evaluation. Evaluation with CCSD(T), which is high-accuracy but high-cost, is an experiment closely aligned with practical applications and likely to attract the readers' interest.

**Other Comments Or Suggestions:**

It would be good to describe how the periodic boundary conditions for bulk were handled when adding noise. Probably, noised atoms outside the unit cell will be wrapped back into the cell.

Typo
* Page 5: "nergy" should be "energy" in Section 5.

**Other Strengths And Weaknesses:**

A strength of this paper is that it provides evaluation results with varying training data, and it consistently demonstrates improved accuracy across various MLIP methods.

**Questions For Authors:**

1. Is the total number of training iterations in Table A2 only for the proposed method? Let me confirm the training iterations are appropriately selected for the baseline methods. Since they are faster to train than the proposed method, they should be able to run longer iterations in the same amount of time.   Did the baseline methods require more iterations to converge compared to the proposed method?
2. It is expected that PISC, being a second-order approximation, would be sensitive to the value of the maximum perturbation length, $\epsilon$. Please provide the performance evaluation results when varying epsilon. Also, how was the 30% of the original bond length determined for epsilon?
3. Did you even use direct force prediction for MACE? As shown in the Stability evaluation results in Figure 3, direct force prediction cannot be used in MD simulations and is known to have a very limited range of applications. Therefore, readers are likely to be more interested in the results of force calculations using gradients.
4. Could you please provide the formula used to calculate the Relative performance gains shown in Figure 2? Wouldn't it be more intuitive and easier to understand if you showed it as $1 - \frac{RMSE_{PIWSL}}{ RMSE_{baseline}}$?
5. Could you provide a more detailed explanation for the annotation regarding "F" in Table 3? Does it mean that the forces calculated using CCSD(T)/CBS were not used, and instead, the DFT forces were used as the ground truth?

**Relation To Broader Scientific Literature:**

Several foundational models for machine learning potentials have been proposed, but these foundational models are often fine-tuned for specific targets. Additionally, there are various methods for first-principle calculations used to generate training data; methods like CCSD(T) that have smaller errors concerning experimental values tend to have high computational costs.
The proposed method is effective for tasks with limited training data, and it is expected to be beneficial for fine-tuning foundational models as well as for applications that use training data generated from high-cost first-principles calculations.

**Theoretical Claims:**

There is no theoretical claim.

---

> ### Author Rebuttal · Authors · 2025-03-31
>
> Dear Reviewer,
>
> Thank you for your constructive feedback. We have carefully considered your comments and will incorporate the necessary revisions in the camera-ready version. Below, we provide detailed responses to each point.
>
> $\\textbf{Q1}$. The iterations in Table A2 apply to both the baseline and PIWSL. Baseline methods were trained until convergence based on standard criteria, ensuring a fair comparison. Table A20 shows that while the baseline overfits with prolonged training, PIWSL continues improving due to physics-informed weak labels, which is especially beneficial in sparse data regimes.
>
> To further support this, we extended MACE-OFF training:
>
> $$\\begin{array} {|r|r|}\\hline  & Model & Epoch & E-RMSE & F-RMSE \\\ \\hline MACE-OFF & Baseline & 100 & 1.21 & 6.90 \\\ \\hline  &  & 1000 & 1.19 & 6.24 \\\ \\hline  \\end{array}$$
>
> These results confirm PIWSL’s effectiveness even when the baseline trains significantly longer. We will include this in the final version.
>
> $\textbf{Q2}$. Below, we present new results on $\\epsilon$ dependence for fine-tuning MACE-OFF, supplementing Table 3:
>
> $$\\begin{array} {|r|r|}\\hline  & Model & \\epsilon & E-RMSE & F-RMSE \\\ \\hline MACE-OFF & PIWSL & 0.005 & 0.73 & 4.20 \\\ \\hline  &  & 0.01 (Table 3) & 0.72 & 3.77 \\\ \\hline  &  & 0.02 & 0.75 & 4.04 \\\ \\hline  \\end{array}$$
>
> These results show that PIWSL maintains stable performance across reasonable $\\epsilon$ variations. The choice of $\\epsilon$ = 0.3 of the original bond length was empirically determined to ensure the validity of the Taylor expansion. Table A18 shows that perturbations closer this threshold degrade performance, indicating the breakdown of the approximation. This aligns with constraints observed in prior studies, which we will clarify in the final version.
>
> $\textbf{Q3}$. No. MACE does not include a direct force module; we used official MACE models. We will explicitly mention this.
>
> $\textbf{Q4}$. The RMSE ratio is computed as RMSE_PIWSL / RMSE_Baseline to determine whether the PIWSL error is lower than that of the baseline. Thank you for your observation—we will add an explicit definition in the final version.
>
> $\textbf{Q5}$. To clarify, CCSD/CBS was not used to generate the energy and force labels in this experiment. Instead, CCSD/cc-pVDZ was used [1]. During training, we relied solely on energy labels, while CCSD/cc-pVDZ force labels were used only in the test dataset to evaluate whether PIWSL could improve force predictions without force labels during training. This setup is motivated by real-world applications where force labels are often unavailable due to the high computational cost of advanced coupled cluster methods, such as CC/CBS. By testing in this way, we assess PIWSL’s ability to generalize force predictions from energy supervision alone.
>
> [1] Chmiela. et al. Nat Commun 9, 3887 (2018).
>
> $\textbf{Comparison with DeNS (Liao+2024)}$
>
> The key differences between PIWSL and DeNS are as follows:
>
> 1) DeNS requires force labels, whereas PIWSL does not. This allows PIWSL to fine-tune MLIP models using datasets containing only potential energies, as demonstrated in Table 3. This scenario is particularly interesting for methods like CC/CBS.
>
> 2) While DeNS introduces a smaller increase in training time, PIWSL offers greater flexibility by eliminating the need for force supervision, making it more broadly applicable in scenarios where force labels are unavailable.
>
> 3) DeNS incorporates an equivariant force encoding module within the target MLIP model, which requires integration into the training pipeline. In contrast, PIWSL directly refines energy-based models without such modifications.
>
> 4) DeNS primarily discusses energy and force errors but does not explicitly discuss robustness in molecular dynamics (MD) simulations, which is one of the key aspects in our work.
>
> Additionally, an interesting open question is whether techniques from DeNS could complement PIWSL to further enhance MLIP performance. While we have not explored this combination in this work, investigating such synergies could be valuable for future research.
> We acknowledge that DeNS was recently accepted on December 27th. Although we have already cited it in Appendix C.2 as a related work, we will provide a more detailed discussion of these differences in the “Related Work” section of the main body in the camera-ready version.
>
> $\textbf{Periodic boundary condition}$
>
> As noted by the reviewer, atoms outside the unit cell are wrapped back into the cell when noise is applied. We will add this clarification to the camera-ready version.
>
> $\textbf{Physics-Informed Claim for PITC}$
>
> Concerning the physical motivation behind PITC: It is valid to locally approximate a potential energy surface using a Taylor series.

---

### Decision · Program_Chairs · 2025-05-01

**Decision:**

Accept (poster)

**Comment:**

Dear authors,

thank you for submitting your paper to this year's ICML conference. After extensive review, rebuttal period and discussion phase, the reviewers agreed that your paper has good novelty to be accepted. Your paper proposed a physics-informed weakly supervised learning (PIWSL) framework that can improve the training of interatomic potentials. Your method incorporates two novel loss functions—PITC and PISC—motivated by physical principles such as energy conservation and Taylor series approximations.

Despite the fact that several weaknesses were initially identified (computational cost, limited diversity of applications, ...) most of the concerns have been sufficiently addressed during rebuttal period.
The extensive empirical results in your paper covers several benchmark datasets (e.g., ANI-1x, $TiO_2$, MD17) and models, showing improved accuracy, robustness, and data efficiency.

I would like to congratulate you for acceptance of your work and strongly encourage you to implement all the suggested improvements and results of discussions into the final camera ready paper.

Best, AC